# AiDE-Q: Synthetic Labeled Datasets Can Enhance Learning Models for Quantum Property Estimation

**Xinbiao Wang**[1,3], **Yuxuan Du**[1,2]*, **Zihan Lou**[3], **Yang Qian**[4], **Kaining Zhang**[1],
**Yong Luo**[3]*, **Bo Du**[3], **Dacheng Tao**[1]*

[1]College of Computing and Data Science,
Nanyang Technological University, Singapore, Singapore
[2]School of Physical and Mathematical Sciences,
Nanyang Technological University, Singapore, Singapore
[3]National Engineering Research Center for Multimedia Software,
School of Computer Science, Wuhan University, Wuhan, China
[4]Department of Data Science, City University of Hong Kong, Hong Kong, Hong Kong
`yuxuan.du@ntu.edu.sg,yluo180@gmail.com,dacheng.tao@ntu.edu.sg`

## Abstract

Quantum many-body problems are central to various scientific disciplines, yet their ground-state properties are intrinsically challenging to estimate. Recent advances in deep learning (DL) offer potential solutions in this field, complementing prior purely classical and quantum approaches. However, existing DL-based models typically assume access to a large-scale and noiseless labeled dataset collected by infinite sampling. This idealization raises fundamental concerns about their practical utility, especially given the limited availability of quantum hardware in the near term. To unleash the power of these DL-based models, we propose AiDE-Q (automatic data engine for quantum property estimation), an effective framework that addresses this challenge by iteratively generating high-quality synthetic labeled datasets. Specifically, AiDE-Q utilizes a consistency-check method to assess the quality of synthetic labels and continuously improves the employed DL models with the identified high-quality synthetic dataset. To verify the effectiveness of AiDE-Q, we conduct extensive numerical simulations on a diverse set of quantum many-body and molecular systems, with up to 50 qubits. The results show that AiDE-Q enhances prediction performance for various reference learning models, with improvements of up to $14.2\%$. Moreover, we exhibit that a basic supervised learning model integrated with AiDE-Q outperforms advanced reference models, highlighting the importance of a synthetic dataset. Our work paves the way for more efficient and practical applications of DL for quantum property estimation.

## 1 Introduction

Many fundamental problems across diverse scientific disciplines, from condensed matter physics to quantum chemistry and materials science, can be reduced to solving quantum many-body problems [1–3]. A central challenge in this regime is characterizing ground-state properties, a task known as quantum property estimation (QPE), which offers critical insight into the behavior of complex quantum systems [4–6]. Numerous methods have been proposed towards QPE, ranging from classical simulations [7–11] to quantum algorithms for state learning [12–17], but their applicability remains limited. That is, classical simulations face exponential computational costs as the system size increases [18, 19], while quantum algorithms often require extensive measurements and complex

---

*Corresponding authors.

39th Conference on Neural Information Processing Systems (NeurIPS 2025).

operations to evaluate intricate properties such as entanglement entropy [20, 21]. These difficulties are further compounded by the scarcity of quantum resources in the early stages of quantum computing.

Learning-based approaches, especially deep learning (DL) algorithms, have recently emerged as promising solutions for QPE [22–39], particularly for systems with shared features. These methods involve training a deep neural network on a large dataset of measurement data obtained from quantum many-body states with varying parameters, then leveraging the trained model to predict the properties of previously unseen quantum states. Over the past years, huge efforts have been devoted to exploring different learning paradigms and neural architectures to achieve accurate property estimation while using fewer measurements. In this endeavor, existing DL models can be broadly categorized into three paradigms, which are *supervised learning* [40–50], *semi-supervised learning* [51], and *self-supervised learning and fine-tuning* [52, 53]. Despite their promise, all of these approaches often assume access to high-quality training datasets with exact labels, overlooking the exponentially increasing overhead

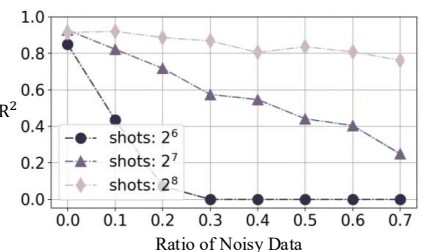

Figure 1: Coefficient of determination $R^2$ of DL models for predicting entanglement entropy of 50-qubit Heisenberg models. The prediction performance decreases as the ratio of noisy labels in training datasets increases or the number of measurements decreases.

of dataset construction as the system size grows. This ignorance poses two severe issues: (i) The reported empirical results fail to reflect the true performance of these methods in real-world scenarios. (ii) How to train these DL models with limited quantum resources remains unknown.

Compared to training on ideal datasets with exact labels, learning under limited quantum resources poses substantially greater challenges for DL models. Specifically, in this scenario, we have three choices for the dataset construction: (i) a dataset consisting of small samples with approximately accurate labels; (ii) a dataset consisting of a large number of samples with noisy labels; (iii) a hybrid dataset consisting of a few samples with approximately accurate labels and numerous samples with noisy labels. Notably, the first two choices are denied by the well-established principle in DL theory, where small-sized datasets lead to overfitting and poor generalization, and a dataset with noisy labels often results in incorrect learning [54–57]. As a viable alternative, hybrid datasets offer a principled compromise, enabling a balance between label accuracy and dataset scale under realistic resource constraints. However, direct training on the hybrid dataset could *significantly degrade* the predictive performance of DL models, as empirical evidence is given in Fig. 1. In this regard, a critical question is: *How to maximally exploit the hybrid dataset to improve the performance of DL models in QPE?*

To address this question, here we propose **AiDE-Q** (**A**utomatic **D**ata **E**ngine for **Q**PE), a *simple but effective* framework inspired by traditional data engines [58–64] that iteratively enhances DL models by acquiring data with high-quality synthetic labels for training. A notable feature of AiDE-Q is its **compatibility** to most DL models for QPE. Without increasing any measurement overhead, AiDE-Q demonstrably enhances the performance of the underlying DL model. On the algorithmic level, our **key technical contribution** is introducing a *consistency-check method* of AiDE-Q to assess the quality of the synthetic labels and select the high-quality data to further train the employed DL model. Unlike traditional data engines [58, 59, 63], which typically rely on extensive manual intervention for labeling and selecting high-quality data, AiDE-Q automates this process by employing a DL model for data labeling and a consistency-check module for data selection. As an **additional contribution**, we systematically conduct extensive numerical experiments for integrating AiDE-Q into various DL paradigms on predicting the entanglement entropy and correlation of the Heisenberg XXZ model and cluster Ising model with up to 50 qubits. The results show that AiDE-Q could effectively improve the prediction performance of various reference DL models, with the highest improvement reaching 14.2%. A notable phenomenon is that *a vanilla supervised learning model integrated with AiDE-Q outperforms the state-of-the-art reference learning models*. This finding underscores the critical role of pioneering synthetic data in advancing the applicability of learning-based methods to QPE tasks. We release the source code and dataset at Github to facilitate future research in this domain.

## 2 Preliminaries

In this section, we outline the essential background on quantum computation, classical shadows, and quantum property estimation (QPE). Additional technical details are provided in Appendix A.

**Basics of quantum computing.** The elementary unit of quantum computation is the *qubit* (or quantum bit) [65], which is the quantum mechanical analog of a classical bit. A qubit is a two-level quantum-mechanical system described by a unit vector in the Hilbert space $\mathbb{C}^2$. In Dirac notation, a qubit state is defined as $|\phi\rangle = c_0 |0\rangle + c_1 |1\rangle \in \mathbb{C}^2$ where $|0\rangle = [1, 0]^\dagger$ and $|1\rangle = [0, 1]^\dagger$ specify two unit bases and the coefficients $c_0, c_1 \in \mathbb{C}$ yield $|c_0|^2 + |c_1|^2 = 1$. Similarly, the *quantum state* of $n$ qubits is defined as a unit vector in $\mathbb{C}^{2^n}$, i.e., $|\psi\rangle = \sum_{j=1}^{2^n} c_j |e_j\rangle$, where $|e_j\rangle \in \mathbb{R}^{2^n}$ is the computational basis whose $j$-th entry is 1 and other entries are 0, and $\sum_{j=1}^{2^n} |c_j|^2 = 1$ with $c_j \in \mathbb{C}$. Besides Dirac notation, the density matrix can be used to describe more general qubit states. For example, the density matrix of the state $|\psi\rangle$ is $\rho = |\psi\rangle\langle\psi| \in \mathbb{C}^{2^n \times 2^n}$, where $\langle\psi| = |\psi\rangle^\dagger$ refers to the complex conjugate transpose of $|\psi\rangle$. For a set of qubit states $\{p_j, |\psi_j\rangle\}_{j=1}^m$ with $p_j > 0$, $\sum_{j=1}^m p_j = 1$, and $|\psi_j\rangle \in \mathbb{C}^{2^n}$ for $j \in [m]$, its density matrix is $\rho = \sum_{j=1}^m p_j \rho_j$ with $\rho_j = |\psi_j\rangle\langle\psi_j|$ and $\mathrm{Tr}(\rho) = 1$.

The *quantum measurement* refers to the procedure of extracting classical information from the quantum state. It is mathematically specified by a Hermitian matrix $H$ called the *observable*. Applying the observable $H$ to the quantum state $|\psi\rangle$ yields a random variable whose expectation value is $\langle\psi| H |\psi\rangle$ or $\mathrm{Tr}(H\rho)$ for the density matrix $\rho = |\psi\rangle\langle\psi|$. For instance, applying computational basis measurement $|0\rangle\langle 0|$ on a quantum state $\rho$ for $m$ times yields a bit string $\boldsymbol{b} \in \{0, 1\}^m$, and the estimated expectation values is given by $\bar{\boldsymbol{b}} = \sum_{i=1}^m \boldsymbol{b}_i/m$ with $\mathbb{E}(\bar{\boldsymbol{b}}) = \mathrm{Tr}(\rho |0\rangle\langle 0|)$.

**Classical shadow.** The classical shadow [13] of a quantum state $\rho$ is constructed using a set of randomized measurements. Given a unitary operator $U$ sampled from a unitary ensemble $\mathcal{U}$, and the subsequent measurement outcome $\boldsymbol{b}$ under computational measurement, the classical shadow from one snapshot is represented by $\hat{\rho} = \mathcal{N}^{-1}(U^\dagger |\boldsymbol{b}\rangle\langle\boldsymbol{b}| U)$, where $\mathcal{N}$ is a linear map associated with the measurement process. In this manuscript, we focus on the random Pauli measurement such that the classical shadow from $m$ snapshots refers to $\hat{\rho} = \frac{1}{m} \sum_{j=1}^m \bigotimes_{i=1}^N (3U_j^{(i)\dagger} |\boldsymbol{b}_j^{(i)}\rangle\langle\boldsymbol{b}_j^{(i)}| U_j^{(i)} - I_2)$, where random unitaries $U_j^{(i)}$ are sampled from the Pauli ensembles $\mathcal{U} = \{I_2, X, Y, Z\}^{\otimes n} \backslash I_2^{\otimes n}$ with $I_2 = \left(\begin{smallmatrix} 1 & 0 \\ 0 & 1 \end{smallmatrix}\right)$, and $X = \left(\begin{smallmatrix} 0 & 1 \\ 1 & 0 \end{smallmatrix}\right)$, $Y = \left(\begin{smallmatrix} 0 & -i \\ i & 0 \end{smallmatrix}\right)$, $Z = \left(\begin{smallmatrix} 1 & 0 \\ 0 & -1 \end{smallmatrix}\right)$) being Pauli-X, -Y, -Z operators.

**Quantum properties estimation (QPE).** We consider the QPE task for ground states of quantum many-body systems with physical parameters $\boldsymbol{p}$, described by a Hermitian $H(\boldsymbol{p})$. The *ground state* $|\psi(\boldsymbol{p})\rangle$ is defined as the eigenvector relating to the minimum eigenvalues of $H(\boldsymbol{p})$. The task of QPE involves predicting properties $f(\rho(\boldsymbol{p}))$ with $\rho(\boldsymbol{p}) = |\psi(\boldsymbol{p})\rangle\langle\psi(\boldsymbol{p})|$ using the measurement outcomes $\boldsymbol{o}$ obtained by applying measurement operators $\boldsymbol{M}$ to the quantum state $\rho(\boldsymbol{p})$.

Two important QPE tasks are estimating the *Renyi entanglement entropy* and *two-point correlations* in quantum many-body systems, which are respectively defined as

$$S_A(\rho) = -\log_2 \mathrm{Tr}(\rho_A^2), \text{ and } \mathcal{C}_{ij}^\alpha(\rho) = \mathrm{Tr}\left(\rho \sigma_i^\alpha \sigma_j^\alpha\right), \tag{1}$$

where $\alpha \in \{z, x\}$, and $\sigma_i^z$ ($\sigma_i^x$) refers to the Pauli-Z (Pauli-X) operator acting on the $i$-th qubit. Here, $A$ refers to the index set of the subsystem and $\rho_A$ is the reduced density matrix of $\rho$ on the subsystem $A$. For two-point correlations, the two indices $i, j$ satisfy $1 \leq i \neq j \leq n$. These properties are widely studied as standard tasks in many studies of quantum states learning [13, 48, 52], and are essential for understanding critical behavior like phase transitions in quantum many-body systems [66, 67].

**Related work.** We highlight that there are no comparative studies with our work, as AiDE-Q is compatible with various DL-based models for QPE. Refer to Appendix B for elaborations.

## 3 Problem setup and implementation of AiDE-Q

For clarity, we recap the mechanism of DL models for QPE training on an ideal dataset in Sec. 3.1. Then, we formulate the learning-based QPE tasks with a limited measurement budget and present the implementation of AiDE-Q and exhibit its compatibility with various DL models in Sec. 3.2.

### 3.1 Learning-based QPE models with an ideal dataset

Existing DL-based models for QPE can be broadly categorized into three learning paradigms: supervised learning (SL), semi-supervised learning (SSL), and self-supervised learning with fine-tuning (SSL-FT). Despite differences in label usage, all three follow a common two-stage pipeline: (i) dataset construction and (ii) model implementation and optimization. In what follows, we first describe the data collection process specific to QPE, and then summarize the learning procedures in each paradigm, emphasizing their differences at each stage. We defer the details to Appendix C.

**Data collection for QPE.** We define the classical data representation by reviewing the process of obtaining classical data through the classical shadow method, as introduced in Sec. 2. Consider an $N$-qubit quantum many-body state $\rho(\boldsymbol{p})$ parameterized by a $d$-dimensional physical parameter $\boldsymbol{p} \in \mathbb{R}^d$, the classical shadow approach involves performing $m$ random Pauli measurements, denoted by $\boldsymbol{M} = (\boldsymbol{M}_1, \cdots, \boldsymbol{M}_m) \in \mathcal{U}^m$ on the quantum state $\rho(\boldsymbol{p})$ with $\mathcal{U}$ being the Pauli ensemble. The measurement outcomes are denoted by $\boldsymbol{o} \in \{0,1\}^{Nm}$, where $\boldsymbol{o}_{ij}$ refers to the outcome of the Pauli operator $\boldsymbol{M}_i$ on the $j$-th qubit. As such, we define the *classical data description* of the quantum state $\rho(\boldsymbol{p})$ as $\boldsymbol{x}(\boldsymbol{p}) = (\boldsymbol{p}, \boldsymbol{M}, \boldsymbol{o})$. Additionally, we denote the exact quantum properties of various subsystems by a *label vector* $\boldsymbol{y}$, where each entry $\boldsymbol{y}_i$ corresponds to a quantum property for a specific subsystem. For example, the label $\boldsymbol{y}_i$ for entanglement entropy of a subsystem $A_i$ in Eq. (1) is given by $S_{A_i}(\rho(\boldsymbol{p}))$. Formally, a classical data point with the ideal label for explored properties for $\rho(\boldsymbol{p})$ is

$$(\boldsymbol{x}(\boldsymbol{p}), \boldsymbol{y}(\boldsymbol{p})) \equiv (\boldsymbol{p}, \boldsymbol{M}(\boldsymbol{p}), \boldsymbol{o}(\boldsymbol{p}), \boldsymbol{y}(\boldsymbol{p})). \tag{2}$$

Through sampling different physical parameters $\boldsymbol{p}$ from a specified distribution, the training dataset for DL models is constructed. Hereafter, for simplicity, we omit the dependence of $\boldsymbol{x}$, $\boldsymbol{M}$, $\boldsymbol{o}$, and $\boldsymbol{y}$ on the parameters $\boldsymbol{p}$ when no ambiguity arises, unless otherwise specified.

**Remark.** The data collection from quantum systems can utilize other measurement operators $\boldsymbol{M}$ beyond random Pauli measurements, such as information-complete measurement operators [68, 69], to collect measurement outputs $\boldsymbol{o}$. Additionally, not all elements in the triplet $(\boldsymbol{p}, \boldsymbol{M}, \boldsymbol{o})$ are necessary for model training, depending on the specific DL model being used. Refer to Appendix C for details.

**SL paradigm.** A large number of studies utilize the SL paradigm to address different QPE tasks, exploring various neural architectures such as multi-layer perceptrons [45, 48] and convolutional neural networks [29, 43, 47, 70], along with different loss functions like mean squared error (MSE) and cross-entropy. In this paradigm, the dataset $\mathcal{S}_{\mathrm{SL}} = \{(\boldsymbol{x}^{(i)}, \boldsymbol{y}^{(i)}))\}_{i=1}^n$ is constructed to train the learning model $f_{\mathrm{SL}}$ under a loss function $\mathcal{L}_{\mathrm{SL}}$, which for MSE is defined as,

$$\mathcal{L}_{\mathrm{SL}} = \frac{1}{n} \sum_{i=1}^n \left\| f_{\mathrm{SL}}(\boldsymbol{x}^{(i)}) - \boldsymbol{y}^{(i)} \right\|^2, \tag{3}$$

where $\boldsymbol{x}^{(i)} = (\boldsymbol{p}^{(i)}, \boldsymbol{M}^{(i)}, \boldsymbol{o}^{(i)})$ are collected from $n$ different quantum states $\rho(\boldsymbol{p}^{(i)})$ with varying parameters $\boldsymbol{p}^{(i)}$ sampled from a specific distribution $\mathcal{D}$. The exact labels $\boldsymbol{y}^{(i)}$ are assumed to be available for all data in $\mathcal{S}_{\mathrm{SL}}$. Finally, the trained model $f_{\mathrm{SL}}$ is used to predict the quantum properties $\boldsymbol{y}'$ based on the collected data $\boldsymbol{x}'$ collected from unseen quantum states $\rho(\boldsymbol{p}')$.

**SSL paradigm.** The SSL paradigm involves training a learning model $f_{\mathrm{SSL}}$ with a dataset consisting of both labeled and unlabeled data, i.e., $\mathcal{S}_{\mathrm{SSL}} = \{(\boldsymbol{x}^{(i)}, \boldsymbol{y}^{(i)})\}_{i=1}^{n_l} \cup \{\boldsymbol{x}^{(i)}\}_{i=n_l+1}^n$, where $n_l$ is the number of labeled data points. The only work in this paradigm is Ref. [51], which exploits a teacher-student architecture to train the learning model $f_{\mathrm{SSL}}$. Specifically, this architecture comprises a student model $f_{\mathrm{SSL}}$ and a teacher model $f'_{\mathrm{SSL}}$, both sharing identical structural designs. The student model learns through both a supervised loss and a consistency loss

$$\mathcal{L}_{\mathrm{SSL}} = \frac{1}{n_l} \sum_{i=1}^{n_l} \left\| f_{\mathrm{SSL}}(\boldsymbol{x}^{(i)}) - \boldsymbol{y}^{(i)} \right\|^2 + \frac{\lambda}{n - n_l} \sum_{i=n_l+1}^n \left\| f_{\mathrm{SSL}}(\boldsymbol{x}^{(i)}) - f'_{\mathrm{SSL}}(\boldsymbol{x}^{(i)}) \right\|^2. \tag{4}$$

where $\lambda$ refers to the consistency weight. The parameters of the teacher model are maintained as an exponential moving average of the student model's parameters throughout training [71]. Once trained, the student model $f_{\mathrm{SSL}}$ is employed to perform the prediction.

**SSL-FT paradigm.** The dataset $\mathcal{S}_{\mathrm{SF}}$ considered in SSL-FT is similar to $\mathcal{S}_{\mathrm{SSL}}$ in SSL, which consists of both labeled and unlabeled data. However, unlike SSL, SSL-FT first performs self-supervised

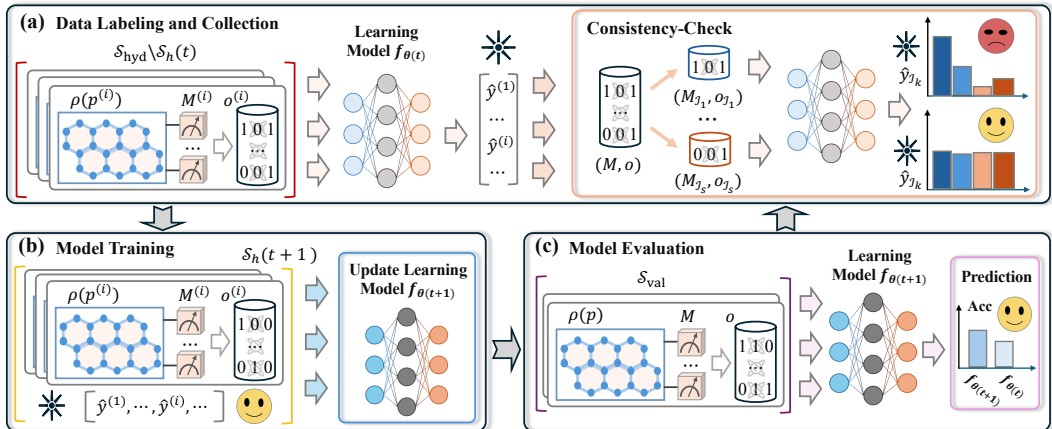

Figure 2: Framewrok of the AiDE-Q. AiDE-Q follows an iterative pipeline consisting of three primary stages: (a) *data labeling and collection*: this stage first use the trained model $f_{\boldsymbol{\theta}(t)}$ at the $t$-iteration to generate labels for the data in $\mathcal{S}_{\mathrm{hyd}}\backslash\mathcal{S}_h(t)$, and then using the consistency-check to collect the data $(p, \boldsymbol{M}, \boldsymbol{o})$ and its synthetic label $\hat{\boldsymbol{y}}$ with small variance among the $s$ generated labels $\hat{\boldsymbol{y}}_{\mathcal{I}_k}$ of the masked data $(\boldsymbol{M}_{\mathcal{I}_k}, \boldsymbol{o}_{\mathcal{I}_k})$, as defined in Eq. (6); (b) *model training*: this stage further fine-tunes the DL model with the updated dataset $\mathcal{S}_h(t+1)$ and obtain a new DL model $f_{\boldsymbol{\theta}(t+1)}$; (c) *model evaluation*: the updated DL model $f_{\boldsymbol{\theta}(t+1)}$ is evaluated on a validation dataset $\mathcal{S}_{\mathrm{val}}$ to examine whether the prediction performance is improved compared to $f_{\boldsymbol{\theta}(t)}$.

learning on the unlabeled dataset to obtain a pre-trained model $f_{\mathrm{SF}}$ that can extract generic and meaningful hidden features, and then fine-tunes $f_{\mathrm{SF}}$ using the labeled data for specific QPE tasks [52, 53]. The loss function for model fine-tuning follows the SL paradigm, as defined in Eq. (3).

Despite the different learning paradigms and implementation details, almost all existing DL models utilize ideal labels $\{\boldsymbol{y}^{(i)}\}$ during training, which are often unavailable due to the exponentially increasing computational overhead to acquire as the size of the quantum system grows. This limitation makes it unclear whether DL algorithms can effectively tackle QPE tasks in practice, where the number of measurements is restricted and the accessible dataset is hybrid.

## 3.2  Implementation of AiDE-Q towards the hybrid dataset

Before we present the implementation details of AiDE-Q, let us first formulate the objective of DL models given a hybrid dataset, where the total number of measurements for data collection is limited.

**DL-based models for QPE with a hybrid dataset.** To account for practical measurement overhead, the label of $\rho(\boldsymbol{p})$ can only be derived from collected measurement data $\boldsymbol{o}(\boldsymbol{p})$ in Eq. (2). As a result, the label may be noisy, with the noise level determined by the number of measurements $m$. Let the noisy label of $\rho(\boldsymbol{p})$ be $\hat{\boldsymbol{y}}$. Without loss of generality, the hybrid dataset takes the form as

$$\mathcal{S}_{\mathrm{hyd}} = \mathcal{S}_L \cup \mathcal{S}_U, \text{ with } \mathcal{S}_L = \{\boldsymbol{x}^{(i)}|\boldsymbol{o}^{(i)} \in \mathbb{R}^{Nm_l}\}_{i=1}^{n_l} \text{ and } \mathcal{S}_U = \{\boldsymbol{x}^{(i)}|\boldsymbol{o}^{(i)} \in \mathbb{R}^{Nm_u}\}_{i=n_l+1}^{n_l+n_u}, \quad (5)$$

where the training dataset $\mathcal{S}_{\mathrm{hyd}}$ consists of $\mathcal{S}_L$ and $\mathcal{S}_U$, in which the number of examples are respectively denoted as $n_l$ and $n_u$. The number of measurements per example in $\mathcal{S}_L$ and $\mathcal{S}_U$ is denoted by $m_l$ and $m_u$ with $m_l > m_u$. Specifically, each example in $\mathcal{S}_L$ contains a large number of measurements, yielding a small difference with the accurate label $\boldsymbol{y}$. In contrast, $\mathcal{S}_U$ contains more data points $n_u$, though at the cost of increased label noise between $\hat{\boldsymbol{y}}$ and $\boldsymbol{y}$.

**Overview of AiDE-Q.** Our initial empirical findings, as shown in Fig. 1, suggest that naïvely training existing DL models on the hybrid dataset $\mathcal{S}_{\mathrm{hyd}}$ can result in suboptimal predictive performance. This underscores the need for novel methods that allow DL-based models for QPE to efficiently learn from hybrid datasets and deliver accurate predictions.

To address this challenge, we propose the automatic data engine for QPE (AiDE-Q), an effective and scalable framework that automatically identifies and generates high-quality data labels through iterative interactions between various learning models and the hybrid dataset, thereby continually enhancing model performance. An overview of AiDE-Q is shown in Fig. 2. Given a learning model $f_{\boldsymbol{\theta}}$ and a hybrid dataset $\mathcal{S}_{\mathrm{hyd}}$ in Eq. (5), AiDE-Q involves iteratively updating a subset of

the training dataset $\mathcal{S}_h \subset \mathcal{S}_{\mathrm{hyd}}$ with high-quality labels and training $f_{\boldsymbol{\theta}}$ on this updated dataset $\mathcal{S}_h$. For clarity, denote $\mathcal{S}_h(t)$ as the high-quality dataset after $t$ updates, and $f_{\boldsymbol{\theta}(t)}$ as the learning model trained on $\mathcal{S}_h(t)$. Initially, we set $\mathcal{S}_h(0) = \mathcal{S}_L$; the employed learning model is flexible, i.e., $f_{\boldsymbol{\theta}(0)} \in \{f_{\mathrm{SL}}, f_{\mathrm{SSL}}, f_{\mathrm{SF}}\}$, which is optimized under the corresponding loss as defined in Eq. (3) and Eq. (4). Once $f_{\boldsymbol{\theta}(0)}$ and $\mathcal{S}_h(0)$ are prepared, AiDE-Q follows an iterative pipeline consisting of three primary stages: *data labeling*, *model training*, and *model evaluation*. In what follows, we detail the procedure of AiDE-Q at the $t$-th update with $0 < t \le T$ and $T$ being the total number of updates.

*Stage I: data labeling.* This stage aims to generate and identify more data with high-quality labels from $\mathcal{S}_U$ using the learning model $f_{\boldsymbol{\theta}}$, and collect the newly identified data $\{(\boldsymbol{x}, \hat{\boldsymbol{y}} = f_{\boldsymbol{\theta}}(\boldsymbol{x}))\}$ into the updated high-quality dataset $\mathcal{S}_h(t + 1)$. More specifically, we define the *data quality* of a data point $(\boldsymbol{x}, \hat{\boldsymbol{y}})$ as the closeness between the synthetic label $\hat{\boldsymbol{y}}$ generated by the learning model $f_{\boldsymbol{\theta}}$ and the ideal label $\boldsymbol{y}$. However, $\boldsymbol{y}$ is generally unavailable due to the prohibitive acquisition cost. In this regard, to evaluate the quality of data with a finite number of measurements, we propose a *consistency-check method* that examines the confidence of $\hat{\boldsymbol{y}}$ as an alternative.

The core idea behind the consistency-check method is to evaluate the variance of synthetic labels generated by $f_{\boldsymbol{\theta}}$ using partial measurement outputs, randomly selected from $(\boldsymbol{M}, \boldsymbol{o})$. As shown in Fig. 2(a), for a given data point $(\boldsymbol{x}, f_{\boldsymbol{\theta}}(\boldsymbol{x}))$ with $\boldsymbol{x} = (\boldsymbol{p}, \boldsymbol{M}, \boldsymbol{o})$ and $\boldsymbol{M}, \boldsymbol{o} \in \mathbb{R}^{Nm}$, we randomly sample $s$ subset $\{\boldsymbol{x}_{\mathcal{I}_k} := (\boldsymbol{p}, \boldsymbol{M}_{\mathcal{I}_k}, \boldsymbol{o}_{\mathcal{I}_k})\}_{k=1}^s$ from the input data $\boldsymbol{x}$, where $\mathcal{I}_k \subset [m]$ refers to the index set uniformly sampled from the set of all column indices of the measurement outputs $\boldsymbol{o} \in \{0, 1\}^{Nm}$. For each subset, we generate the synthetic labels with $\hat{\boldsymbol{y}}_{\mathcal{I}_k} = f_{\boldsymbol{\theta}}(\boldsymbol{x}_{\mathcal{I}_k})$. We call $(\boldsymbol{x}_{\mathcal{I}_k}, \boldsymbol{y}_{\mathcal{I}_k})$ the *masked data* corresponding to $(\boldsymbol{x}, f_{\boldsymbol{\theta}}(\boldsymbol{x}))$. Subsequently, we examine the consistency level of $f_{\boldsymbol{\theta}(t)}(\boldsymbol{x})$ by computing the variance over the $s$ masked estimations $\hat{\boldsymbol{y}}_{\mathcal{I}_k}$, i.e.,

$$\mathrm{Var}(\hat{\boldsymbol{y}}) = \frac{1}{s} \sum_{k=1}^s \|\hat{\boldsymbol{y}}_{\mathcal{I}_k} - \bar{\boldsymbol{y}}\|^2, \text{ with } \bar{\boldsymbol{y}} = \frac{1}{s} \sum_{k=1}^s \hat{\boldsymbol{y}}_{\mathcal{I}_k}. \tag{6}$$

The synthetic labels $\hat{\boldsymbol{y}} = f_{\boldsymbol{\theta}}(\boldsymbol{x})$ estimated with a large number of measurements $m$ or a powerful learning model $f_{\boldsymbol{\theta}}$ lead to an accurate approximation to $\boldsymbol{y}$ with a small variance $\mathrm{Var}(\hat{\boldsymbol{y}})$, and hence could be regard as high-quality data.

Supported by the proposed consistency-check method, AiDE-Q assess $\mathrm{Var}(\hat{\boldsymbol{y}})$ of all training examples $\boldsymbol{x} \in \mathcal{S}_{\mathrm{hyd}} \backslash \mathcal{S}_h(t)$. Those training examples whose $\mathrm{Var}(\hat{\boldsymbol{y}})$ is less than a given threshold $\tau$ are identified as the high-quality labeled data and incorporated into the updated high-quality dataset $\mathcal{S}_h(t + 1)$.

*Stage II: model training.* After Stage I, the learning model $f_{\boldsymbol{\theta}(t)}$ will be further trained on the updated high-quality dataset $\mathcal{S}_h(t + 1)$, yielding an updated learning model $f_{\boldsymbol{\theta}(t+1)}$. In particular, when $t = 0$, any learning paradigm introduced in Sec. 3.1 could be employed to train the learning model $f_{\boldsymbol{\theta}(0)}$ on the whole dataset $\mathcal{S}_{\mathrm{hyd}}$. As shown in Fig. 2(b), when $t > 1$, the learning model $f_{\boldsymbol{\theta}(t)}$ is optimized on the updated high-quality dataset $\mathcal{S}_h(t + 1)$ by minimizing the loss function

$$\mathcal{L}(\boldsymbol{\theta}) = \frac{1}{|\mathcal{S}_h(t + 1)|} \sum_{\boldsymbol{x} \in \mathcal{S}_h(t+1)} \|f_{\boldsymbol{\theta}}(\boldsymbol{x}) - \hat{\boldsymbol{y}}\|^2, \tag{7}$$

where $|\mathcal{S}_h(t + 1)|$ refers to the size of dataset $\mathcal{S}_h(t + 1)$ and $\hat{\boldsymbol{y}}$ denotes the the synthetic labels in $\mathcal{S}_h(t + 1)$. Here, the optimized parameters $\boldsymbol{\theta}(t)$ in the model $f_{\boldsymbol{\theta}(t)}$ serve as the initial parameters for model training of $f_{\boldsymbol{\theta}}$ on the dataset $\mathcal{S}_h(t + 1)$. Throughout $T$ iterations, the neural architecture in $f_{\boldsymbol{\theta}}$ remains unchanged, while only the parameters $\boldsymbol{\theta}$ are updated.

*Stage III: model evaluation.* At shown in Fig. 2(c), the updated $f_{\boldsymbol{\theta}(t+1)}$ obtained from Stage II is evaluated on a validation dataset $\mathcal{S}_{\mathrm{val}}$ to assess whether its predictive performance has improved relative to the previous model $f_{\boldsymbol{\theta}(t)}$. This evaluation determines whether the updated model $f_{\boldsymbol{\theta}(t+1)}$ should be adopted for subsequent iterations. The employed validation dataset consists of quantum data $\boldsymbol{x}$ with the same number of measurements as the data in $\mathcal{S}_U$, i.e., $m = m_u$, while the corresponding label $\hat{\boldsymbol{y}}$ is estimated by a larger number of measurement outcomes with $m = m_l$. If the performance of $f_{\boldsymbol{\theta}(t+1)}$ decreases compared to $f_{\boldsymbol{\theta}(t)}$, the newly added high-quality data in $\mathcal{S}_h(t)$ may contain erroneous or harmful samples. In this case, AiDE-Q raises the threshold $\tau$ in the consistency-check module, re-initiating the high-quality labeled data collection, model training, and evaluation process until the model's performance on the validation dataset $\mathcal{S}_{\mathrm{val}}$ improves (Refer to Appendix C.3 for details). This ensures that only reliable and informative samples that positively contribute to model generalization are included in the updated high-quality dataset $\mathcal{S}_h(t + 1)$.

Table 1: $R^2$ in predicting the entanglement entropy $S_A$, two-point correlations $\mathcal{C}_{1j}^x$ and $\mathcal{C}_{1j}^z$ of 10-qubit XXZ model, where $A = [j]$ and $j \in [N-1]$. The number of measurements for low-quality data is set as $m_u = 2^6$. The best results are emphasized in blue while the second-best results are distinguished in orange.

| Method | $S_A$ | | | $\mathcal{C}_{1j}^x$ | | | $\mathcal{C}_{1j}^z$ | | |
|---|---|---|---|---|---|---|---|---|---|
| | $r = 0.4$ | $r = 0.6$ | $r = 0.8$ | $r = 0.4$ | $r = 0.6$ | $r = 0.8$ | $r = 0.4$ | $r = 0.6$ | $r = 0.8$ |
| SL | 0.722 | 0.740 | 0.820 | 0.848 | 0.883 | 0.930 | 0.915 | 0.936 | 0.953 |
| SSL4Q | 0.823 | 0.804 | 0.814 | 0.938 | 0.951 | 0.951 | 0.958 | 0.966 | 0.958 |
| LLM4QPE | 0.745 | 0.814 | 0.857 | 0.851 | 0.891 | 0.937 | 0.923 | 0.933 | 0.961 |
| NTK | - | - | - | 0.864 | 0.799 | 0.910 | 0.966 | 0.267 | 0.901 |
| CS | - | - | - | 0.308 | 0.308 | 0.308 | 0.605 | 0.605 | 0.605 |
| SL w. DE | 0.825 | 0.864 | 0.904 | 0.944 | 0.972 | 0.985 | 0.966 | 0.978 | 0.989 |

**Remark.** The AiDE-Q framework is flexible and can be integrated into other learning models, such as machine learning models [31, 36]. Refer to Appendix D for the discussion.

## 4 Experiments

In this section, we conduct extensive studies about the effectiveness of AiDE-Q on two standard quantum systems: the Heisenberg XXZ model and the one-dimensional cluster-Ising model. Refer to Appendix F for the omitted details and more results.

### 4.1 Data constructions for the explored quantum systems

**Heisenberg XXZ model.** An $N$-qubit Heisenberg XXZ model is defined by the Hamiltonian [72]

$$H = J \sum_{i=1}^{N/2} (\sigma_{2i-1}^x \sigma_{2i}^x + \sigma_{2i-1}^y \sigma_{2i}^y + \sigma_{2i-1}^z \sigma_{2i}^z) + J' \sum_{i=1}^{N/2-1} (\sigma_{2i}^x \sigma_{2i+1}^x + \sigma_{2i}^y \sigma_{2i+1}^y + \sigma_{2i}^z \sigma_{2i+1}^z), \quad (8)$$

where $J$ and $J'$ refer to the alternating nearest-neighbor spin couplings, and $\sigma_i^x$ ($\sigma_i^{(z)}$) represent the Pauli-X (Pauli-Z) operator acting on the $i$-th qubit. The focus on the Heisenberg model is because it is an important statistical mechanical model used to study the behaviors of magnetic systems [73].

We consider a set of ground states corresponding to $n$ uniformly sampled coupling parameters from the region of $J/J' \in (0, 2)$. The system size $N$ is varied as $N \in \{10, 20, 30, 40, 50\}$, with the corresponding number of sampled parameters for each system size being $n \in \{720, 1280, 1860, 2740\}$. For the hybrid dataset in Eq. (5), the ratio of the number of high-quality data $n_l$ to the total dataset size $n$ is defined as $r := n_l/n \in \{0.2, 0.4, 0.6, 0.8\}$. The number of measurements for the initial high-quality dataset $\mathcal{S}_L$ is set to $m_l = 2^{10}$, while for the initial low-quality dataset $\mathcal{S}_U$, the number of measurements is $m_u \in \{2^5, 2^6, 2^7, 2^8, 2^9\}$. The validation dataset consists of $n_{\text{val}} = 120$ data points, with the number of measurements set to $m_{\text{val}} = m_u$.

**Cluster Ising model.** The $N$-qubit cluster Ising model [74], parameterized by a two-dimensional vector $(h_1, h_2)$, is defined by the Hamiltonian

$$H_{\text{cI}} = -\sum_{i=1}^{N-2} \sigma_i^z \sigma_{i+1}^x \sigma_{i+2}^z - h_1 \sum_{i=1}^{N} \sigma_i^x - h_2 \sum_{i=1}^{N-1} \sigma_i^x \sigma_{i+1}^x. \quad (9)$$

This system has been extensively applied to describe a variety of quantum systems and is closely related to combinatorial optimization problems [75–77]. For the hybrid dataset construction, we uniformly sample $n$ different parameter values in the region $h_1/h_2 \in (0, 2)$. The number of qubits and the training dataset size are set to $N = 9$ and $n = 720$, respectively. Other hyperparameters used to build the hybrid dataset in Eq. (2) are set to be the same as those used in Heisenberg XXZ models.

**QPE tasks.** We focus on the non-linear property, entanglement entropy $S_A$, and linear properties, two-point correlations $\mathcal{C}_{1j}^x, \mathcal{C}_{1j}^z$, as defined in Eq. (1). Here, the subsystem $A$ is defined as $A = [1, \cdots, j]$ and the index $j$ ranges in $\{2, \cdots, N\}$. In this regard, the data label $\boldsymbol{y}$ or the synthetic label $\hat{\boldsymbol{y}}$, which corresponds to the properties of interest for a given input $\boldsymbol{x}$, is an $(N-1)$-dimensional vector.

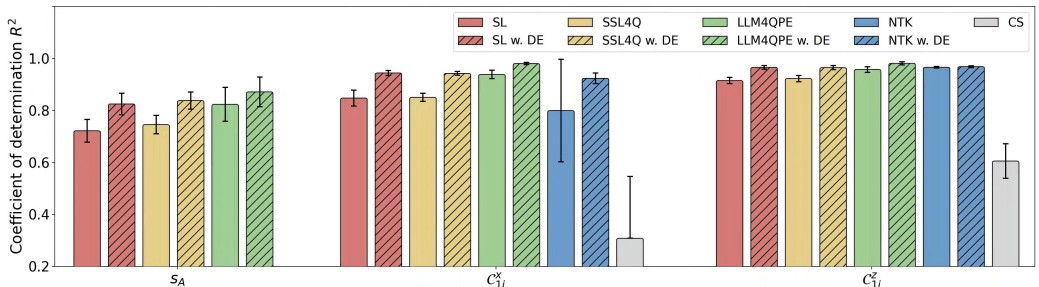

Figure 3: $R^2$ of reference models with and without integrating AiDE-Q in predicting entanglement entropy $S_A$, two-point correlations $\mathcal{C}_{1j}^x$ and $\mathcal{C}_{1j}^z$ of 10-qubit XXZ model, where $A = [j]$ and $j \in [N-1]$. The initial ratio of high-quality data and the number of measurements for low-quality data is set as $r = 0.4$ and $m_u = 2^6$.

## 4.2 Experimental settings

**Reference models.** The first reference model is classical shadow (CS) [13], a learning-free protocol for efficiently predicting various properties of quantum states. For machine-learning-based baselines, we include the neural tangent kernel (NTK) [31] as a representative classical model. Among DL-based models for QPE tasks, we adopt SSL4Q as the reference for the SSL paradigm [51] and LLM4QPE for the SF paradigm [52]. To evaluate AiDE-Q's predictive capabilities under the SL paradigm, we benchmark it as a standalone SL model. All DL models employ identical neural network architectures for feature representations to ensure a fair comparison. Refer to Appendix F for details.

**Evalutaion metrics.** To assess the predictive performance of different learning models, we use the coefficient of determination ($R^2$) as the evaluation metric. Specifically, given the test dataset with ground-truth labels $\{(\boldsymbol{x}^{(i)}, \boldsymbol{y}^{(i)})\}_{i=1}^{n_{\text{te}}}$ of size $n_{\text{te}}$, the coefficient of determination is defined as

$$R^2 = 1 - \frac{\sum_{i=1}^{n_{\text{te}}} (\boldsymbol{y}^{(i)} - f(\boldsymbol{x}^{(i)}))^2}{\sum_{i=1}^{n_{\text{te}}} (\boldsymbol{y}^{(i)} - \bar{\boldsymbol{y}})^2}, \tag{10}$$

where $\bar{\boldsymbol{y}} = \sum_{i=1}^{n_{\text{te}}} \boldsymbol{y}^{(i)}/n_{\text{te}}$ refers to the mean of the true values of the property of interest in the training dataset, and $f(\boldsymbol{x}^{(i)})$ denotes the predicted values from the employed learning model. The quantity $R^2$ typically ranges from 0 to 1, with larger $R^2$ indicating that the model achieves better predictive accuracy. Specifically, when the model's predictions perfectly match the true values, we have $\sum_{i=1}^{n_{\text{te}}} (\boldsymbol{y}^{(i)} - f(\boldsymbol{x}^{(i)}))^2 = 0$ and $R^2 = 1$. This metric eliminates the influence of the magnitude of estimated properties, providing a clearer assessment of a model's predictive ability.

**Hyperparameter settings of AiDE-Q.** The maximum number of iterations of AiDE-Q is set to $T = 6$. For consistency-checking in each iteration, the measurement data is divided into $s = 5$ subsets, each containing $25\%$ of the total number of measurements. Data points whose consistency levels for the synthetic labels fall within the top $10\%$ are used to retrain the learning models.

## 4.3 Experimental results

Despite the wide range of possible hyperparameter configurations, we present numerical results for selected settings to demonstrate AiDE-Q's predictive capabilities when integrated with various learning models, with more numerical results provided in Appendix F.

**Supervised learning models integrated with AiDE-Q outperform all reference models.** Table 1 presents the coefficient of determination $R^2$ for predicting entanglement entropy and two-point correlations for the 10-qubit Heisenberg XXZ model. The results are shown for both machine learning and DL models, with varying ratios of the initial high-quality dataset $\mathcal{S}_L$, while the number of measurements is fixed at $m_u = 2^6$. It can be observed that although DL models trained under the SSL (SSL4Q) and SSL-FT (LLM4QPE) paradigm outperform those trained under the SL paradigm by utilizing the dataset $\mathcal{S}_U$ as unlabeled dataset during training, the SL model incorporated with AiDE-Q demonstrate significantly performance improvement, and consistently achieve superior prediction accuracy across different properties and various ratio settings, as evidenced by their higher $R^2$. The most substantial improvement occurs when predicting entanglement entropy with $r = 0.6$,

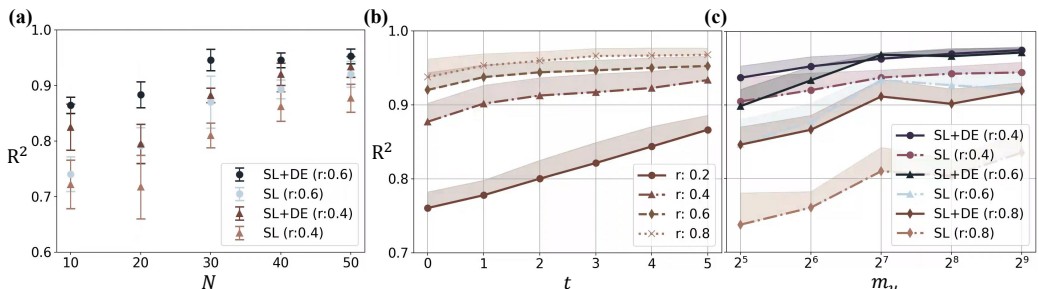

Figure 4: $R^2$ in entanglement entropy prediction for the Heisenberg XXZ model. (a) $R^2$ values of AiDE-Q-integrated SL models across varying quantum system sizes $N$ with different total training dataset sizes and fixed $m_u = 2^6$. (b) Evolution of $R^2$ across AiDE-Q's iterations for 50-qubit XXZ model and fixed $m_u = 2^6$. (c) $R^2$ values with a varying number of measurements $m_u$ for low-quality data points in 50-qubit XXZ model.

Table 2: $R^2$ in predicting the entanglement entropy $S_A$ of 9-qubit cluster Ising models with $A = [j]$ and $j \in [N-1]$. The best results are emphasized in blue while the second-best results are distinguished in orange.

| Method | $m_u = 2^6$ | | | $m_u = 2^7$ | | | $m_u = 2^8$ | | |
|---|---|---|---|---|---|---|---|---|---|
| | $r = 0.4$ | $r = 0.6$ | $r = 0.8$ | $r = 0.4$ | $r = 0.6$ | $r = 0.8$ | $r = 0.4$ | $r = 0.6$ | $r = 0.8$ |
| SL | 0.226 | 0.356 | 0.51 | 0.255 | 0.234 | 0.559 | 0.317 | 0.444 | 0.443 |
| SSL4Q | 0.422 | 0.448 | 0.489 | 0.501 | 0.539 | 0.626 | 0.513 | 0.58 | 0.616 |
| LLM4QPE | 0.29 | 0.322 | 0.436 | 0.322 | 0.308 | 0.587 | 0.194 | 0.458 | 0.458 |
| SL w. DE | 0.563 | 0.791 | 0.873 | 0.697 | 0.788 | 0.908 | 0.686 | 0.83 | 0.896 |
| SSL4Q w. DE | 0.794 | 0.836 | 0.86 | 0.829 | 0.865 | 0.892 | 0.821 | 0.902 | 0.907 |
| LLM4QPE w. DE | 0.569 | 0.767 | 0.901 | 0.596 | 0.821 | 0.939 | 0.646 | 0.842 | 0.924 |

where the SL model incorporated with AiDE-Q increases the $R^2$ from $0.74$ to $0.864$ compared to the vanilla SL model. Notably, this surpasses the prediction performance of the optimal reference model, LLM4QPE, which only achieves an $R^2$ of $0.857$ even with a higher high-quality data ratio $r = 0.8$.

**AiDE-Q enhances the prediction performance of various DL-based models.** We further examine the $R^2$ values of various learning models integrated with AiDE-Q in predicting quantum properties. Fig. 3 presents numerical results with setting $r = 0.4$ and $m_u = 2^6$. The results demonstrate that all learning models show improved $R^2$ after integrating the AiDE-Q into their vanilla learning models. Notably, the improvement is inversely proportional to the original performance—models with lower baseline $R^2$ values show more substantial gains after AiDE-Q incorporation. For example, when predicting two-point correlations $c_{ij}^z$, the $R^2$ values for LLM4QPE, SSL4Q, and NTK methods are improved by $0.042$, $0.092$, and $0.124$ from their original $R^2$ values $0.938$, $0.851$, and $0.799$, repspectively. These findings suggest that AiDE-Q is *compatible* with various learning paradigms, enhancing their prediction performance.

**The scalability of AiDE-Q.** We next evaluate the scalability of AiDE-Q by applying it to the task of predicting entanglement entropy across varying system sizes $N \in \{10, 20, 30, 40, 50\}$ and fixed measurement number $m_u = 2^6$. The achieved results are shown in Fig. 4(a), demonstrating that incorporating AiDE-Q into the SL paradigm consistently enhances prediction performance across all quantum system sizes and initial high-quality data ratios, as evidenced by increased $R^2$ values. These results confirm AiDE-Q's scalability to larger many-body physics models.

**AiDE-Q improves the prediction performance with each iteration.** Fig. 4(b) shows the $R^2$ values for entanglement entropy prediction across iterations of the AiDE-Q under the SL paradigm for the 50-qubit XXZ model. With varying initial ratios of high-quality data $r \in \{0.2, 0.4, 0.6, 0.8\}$ and fixed measurement number $m_u = 2^6$, the $R^2$ values consistently improve as AiDE-Q iterations progress, demonstrating AiDE-Q's effectiveness in identifying high-quality data with a synthetic label to enhance model generalization. Notably, smaller initial rations of high-quality data yield larger performance improvements. For instance, at $r = 0.2$, AiDE-Q constantly increases the $R^2$ from $0.76$ to $0.87$, achieving an improvement of $0.11$, while the improvement is less than $0.07$ for other ratios.

**The effect of the number of measurements on the AiDE-Q.** Fig. 4(c) shows the $R^2$ values for entanglement entropy prediction, evaluated across different numbers of measurements for low-quality

data, $m_u \in \{2^5, \cdots, 2^9\}$, using SL models for the 50-qubit XXZ model. The numerical results demonstrate that increasing the measurement count $m_u$ for the low-quality dataset $\mathcal{S}_U$ effectively enhances the predictive performance of SL models, both with and without AiDE-Q integration.

**Experimental results on the cluster Ising Model.** Table 2 shows the coefficient of determination $R^2$ of entanglement entropy prediction for the 9-qubit cluster Ising model. The results show that integrating AiDE-Q into different learning-based models could significantly improve their predictive performance for different ratios of high-quality data $r \in \{0.4, 0.6, 0.8\}$ and measurement counts $m_u \in \{2^6, 2^7, 2^8\}$. This confirms AiDE-Q's effectiveness across diverse quantum systems.

## 5    Conclusion

In this study, we introduced AiDE-Q, a simple but effective framework that can be integrated into various learning paradigms for quantum property estimation (QPE) to improve the prediction performance of deep learning (DL) models, with limited quantum resources for dataset construction. Numerical experiments on the Heisenberg XXZ and cluster Ising models demonstrate AiDE-Q's effectiveness in enhancing the performance of several learning-based models, highlighting AiDE-Q's potential to improve the practicality of DL for QPE. However, a key limitation of our work is the unknown optimal allocation of limited quantum resources for constructing hybrid datasets.

## Acknowledgment

Y. D. acknowledges the support from the SUG grant of NTU. Y. L. acknowledges the support from National Natural Science Foundation of China (Grant No. U23A20318 and 62276195).

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

# A  More basics of quantum computing

This section provides more details about quantum computing and reviews the classical shadow algorithms for quantum properties estimation (QPE), along with their associated sample complexity.

**Basics of quantum computation.** The elementary unit of quantum computation is the qubit (or quantum bit), which is the quantum mechanical analog of a classical bit. A qubit is a two-level quantum-mechanical system described by a unit vector in the Hilbert space $\mathbb{C}^2$. In Dirac notation, a qubit state is defined as $|\phi\rangle = c_0 |0\rangle + c_1 |1\rangle \in \mathbb{C}^2$ where $|0\rangle = [1, 0]^\top$ and $|1\rangle = [0, 1]^T$ specify two unit bases and the coefficients $c_0, c_1 \in \mathbb{C}$ yield $|c_0|^2 + |c_1|^2 = 1$. Similarly, the *quantum state* of $n$ qubits is defined as a unit vector in $\mathbb{C}^{2^n}$, i.e., $|\psi\rangle = \sum_{j=1}^{2^n} c_j |e_j\rangle$, where $|e_j\rangle \in \mathbb{R}^{2^n}$ is the computational basis whose $j$-th entry is 1 and other entries are 0, and $\sum_{j=1}^{2^n} |c_j|^2 = 1$ with $c_j \in \mathbb{C}$. Besides Dirac notation, the density matrix can be used to describe more general qubit states. For example, the density matrix of the state $|\psi\rangle$ is $\rho = |\psi\rangle \langle\psi| \in \mathbb{C}^{2^n \times 2^n}$, where $\langle\psi| = |\psi\rangle^\dagger$ refers to the complex conjugate transpose of $|\psi\rangle$. For a set of qubit states $\{p_j, |\psi_j\rangle\}_{j=1}^m$ with $p_j > 0$, $\sum_{j=1}^m p_j = 1$, and $|\psi_j\rangle \in \mathbb{C}^{2^n}$ for $j \in [m]$, its density matrix is $\rho = \sum_{j=1}^m p_j \rho_j$ with $\rho_j = |\psi_j\rangle \langle\psi_j|$ and $\text{Tr}(\rho) = 1$.

A *quantum gate* is a unitary operator that can evolve a quantum state $\rho$ to another quantum state $\rho'$. Namely, an $n$-qubit gate $U \in \mathcal{U}(2^n)$ obeys $UU^\dagger = U^\dagger U = I_{2^n}$, where $\mathcal{U}(2^n)$ refers to the unitary group in dimension $2^n$. Typical single-qubit quantum gates include the Pauli gates, which can be written as Pauli matrices:

$$X = \begin{bmatrix} 0 & 1 \\ 1 & 0 \end{bmatrix}, \quad Y = \begin{bmatrix} 0 & -i \\ i & 0 \end{bmatrix}, \quad Z = \begin{bmatrix} 1 & 0 \\ 0 & -1 \end{bmatrix}. \tag{11}$$

The more general quantum gates are their corresponding rotation gates $R_X(\theta) = e^{-i\frac{\theta}{2}X}, R_Y(\theta) = e^{-i\frac{\theta}{2}Y}$, and $R_Z(\theta) = e^{-i\frac{\theta}{2}Z}$ with a tunable parameter $\theta$, which can be written in the matrix form as

$$R_X(\theta) = \begin{bmatrix} \cos\frac{\theta}{2} & -i\sin\frac{\theta}{2} \\ -i\sin\frac{\theta}{2} & \cos\frac{\theta}{2} \end{bmatrix}, R_Y(\theta) = \begin{bmatrix} \cos\frac{\theta}{2} & -\sin\frac{\theta}{2} \\ \sin\frac{\theta}{2} & \cos\frac{\theta}{2} \end{bmatrix}, R_Z(\theta) = \begin{bmatrix} e^{-i\frac{\theta}{2}} & 0 \\ 0 & e^{i\frac{\theta}{2}} \end{bmatrix}. \tag{12}$$

They are equivalent to rotating a tunable angle $\theta$ around $x, y$, and $z$ axes of the Bloch sphere, and recovering the Pauli gates $X, Y$, and $Z$ when $\theta = \pi$. Moreover, a multi-qubit gate can be either an individual gate (e.g., CNOT gate) or a tensor product of multiple single-qubit gates.

The *quantum measurement* refers to the procedure of extracting classical information from the quantum state. It is mathematically specified by a Hermitian matrix $H$ called the *observable*. Applying the observable $H$ to the quantum state $|\psi\rangle$ yields a random variable whose expectation value is $\langle\psi| H |\psi\rangle$.

**Hamiltonian and ground state**. In quantum computation, a *Hamiltonian* is a Hermitian matrix that is used to characterize the evolution of a quantum system or as an observable to extract the classical information from the quantum system. Specifically, under the Schrödinger equation, a quantum gate has the mathematical form of $U = e^{-itH}$, where $H$ is a Hermitian matrix, called the Hamiltonian of the quantum system, and $t$ refers to the evolution time of the Hamiltonian. Typical single-qubit Hamiltonians include the Pauli matrices defined in Eqn. (11). As a result, the evolution time $t$ refers to the tunable parameter $\theta$ in Eqn. (12). Any single-qubit Hamiltonian can be decomposed as the linear combination of Pauli matrices, i.e., $H = a_1 I + a_2 X + a_3 Y + a_4 Z$ with $a_j \in \mathbb{C}$. In the same way, a multi-qubit Hamiltonian is denoted by $H = \sum_{j=1}^{4^n} a_j P_j$, where $P_j \in \{I, X, Y, Z\}^{\otimes n}$ is the tensor product of Pauli matrices. In quantum chemistry and quantum many-body physics, the Hermitian matrix that describes the quantum system to be solved is denoted as the *problem Hamiltonian $H_C$*.

When taking the problem Hamiltonian as the observable, the quantum state $|\psi^*\rangle$ is said to be the *ground state* of problem Hamiltonian $H$ if the expectation value $\langle\psi^*| H |\psi^*\rangle$ takes the minimum eigenvalue of $H$, which is called the *ground energy*. The ground states encode much essential information about the problem Hamiltonian, such as the critical behavior of quantum many-body systems, or the optimal solution of an optimization problem related to the problem Hamiltonian.

Numerous classical and quantum algorithms have been developed to efficiently obtain the ground states of problem Hamiltonians. These algorithms leverage various techniques, including variational

methods, quantum annealing, and the application of tensor networks, to approximate or directly compute the ground state. In particular, quantum algorithms such as the variational quantum eigensolver [78], quantum approximate optimization algorithm [79–82], and the adiabatic quantum algorithm [83] show promising results for solving combinatorial optimization problems by preparing and measuring the ground state of problem Hamiltonians. The efficiency and feasibility of these methods continue to be the subject of extensive research, particularly in the context of near-term quantum devices.

**Classical shadow for QPE.** We review the classical shadow algorithm [13] and the sample complexity for estimating the linear and nonlinear properties of quantum states. Given a unitary operator $U$ sampled from a unitary ensemble $\mathcal{U}$, and the subsequent measurement outcome $\boldsymbol{b}$ under computational measurement, the classical shadow of the $N$-qubit quantum state $\rho$ from $m$ snapshots refers to

$$\hat{\rho} = \frac{1}{m} \sum_{j=1}^{m} \bigotimes_{i=1}^{N} (3U_j^{(i)\dagger} |\boldsymbol{b}_j^{(i)}\rangle \langle \boldsymbol{b}_j^{(i)}| U_j^{(i)} - I_2), \tag{13}$$

where random unitaries $U_j^{(i)}$ are sampled from the Pauli ensembles $\mathcal{U} = \{I_2, X, Y, Z\}^{\otimes n} \backslash I_2^{\otimes n}$. It has been shown that the sample complexity for estimating the expectation of observables $O$ to $\epsilon$-precision is $\mathcal{O}(4^{\text{locality}(O)} \|O\|_\infty^2 / \epsilon^2)$, with locality$(O)$ representing the number of non-identity operators acting on the qubits. For two-point correlation, the observable refers to $O = \sigma_i^x \sigma_j^x$ acting on the $i$-th and $j$-th qubits with locality$(O) = 2$, leading to the sample complexity $\mathcal{O}(16/\epsilon^2)$ for QPE.

The classical shadow could also be used to estimate the entanglement entropy of the quantum state $\rho$ on the subsystem $A$, which is given by

$$\hat{S}_A(\rho) = -\log_2 \hat{\mathcal{P}}(\rho_A), \quad \text{with } \hat{\mathcal{P}}(\rho) = \frac{1}{m(m-1)} \sum_{j \neq j'} \text{Tr}(\hat{\rho}_A^{(j)}) \hat{\rho}_A^{(j')}), \tag{14}$$

where $\hat{\mathcal{P}}(\rho_A)$ refers to the purity estimation of $\text{Tr}(\rho_A^2)$, $\hat{\rho}_A^{(j)} = \bigotimes_{i \in A}^{N} (3U_j^{(i)\dagger} |\boldsymbol{b}_j^{(i)}\rangle \langle \boldsymbol{b}_j^{(i)}| U_j^{(i)} - I_2)$ refers to the local classical shadow on subsystems $A$ obtained from the $j$-th snapshot. It has been shown that the statistical error associated with $\hat{\mathcal{P}}(\rho_A)$ is quantified by its variance, which can be bounded as follows [84, 85],

$$\text{Var}(\hat{\mathcal{P}}(\rho_A)) \leq 4 \left( \frac{2^{|A|} \hat{\mathcal{P}}(\rho_A)}{m} \right) + 2 \left( \frac{2^{2|A|}}{m-1} \right)^2, \tag{15}$$

where $|A|$ denotes the number of qubits in $A$. This bound is known to be essentially optimal [84]. It implies that estimating the entanglement entropy requires an exponentially large number of measurements as the size of subsystem $A$ increases.

# B Related work

In this section, we present a concise review of the literature on quantum property estimation (QPE), categorizing the approaches into three primary paradigms: conventional algorithms, machine learning (ML) algorithms, and deep learning (DL) algorithms. Our discussion emphasizes that the proposed AiDE-Q framework can be seamlessly integrated with various ML and DL algorithms.

**Conventional algorithms for QPE.** Conventional algorithms primarily focus on estimating the quantum properties of individual quantum states. A wide range of methods have been proposed, spanning classical simulation algorithms, quantum algorithms, and quantum state learning (QSL) algorithms [86].

Classical simulation algorithms achieve QPE entirely on classical computers, often utilizing tensor network techniques to simulate the whole quantum state [7–11], or employing Pauli-path simulation methods to estimate properties of quantum circuit generated states [87–91]. However, these classical simulation algorithms are typically limited to specific types of quantum states, such as low-entangled or low-magic states. Quantum algorithms for QPE involve the implementation of well-designed quantum circuits to evolve quantum states and extract the desired properties. Despite their potential, these quantum algorithms often require substantial quantum resources to construct the circuits, which can hinder their practical application, especially in the early stages of quantum computing [92, 93].

Quantum state learning (QSL) algorithms for QPE operate by performing multiple measurements on a quantum state and post-processing the measurement results to estimate the quantum properties of interest [13, 20, 16, 84, 85, 94]. Among these, the classical shadow algorithm has emerged as one of the most popular and efficient approaches, offering rigorous theoretical guarantees [13]. By using the measurement outputs from random measurement operators, this algorithm can simultaneously estimate multiple properties of quantum states, making it one of the most resource-efficient conventional algorithms for QPE.

**ML algorithms for QPE.** Unlike conventional algorithms that focus on QPE of individual states, machine learning algorithms address the QPE problem for classes of quantum states originating from the same quantum many-body systems with varying physical parameters. Notably, Huang et al. [95] proposed a kernel-based learning model that efficiently predicts linear properties of quantum many-body states with rigorous theoretical guarantees. This method eliminates the need for quantum devices during the prediction phase. Furthermore, Lewis et al. [36] introduced a Lasso regression model for $N$-qubit gapped local Hamiltonians that improves sample complexity from polynomial scaling $N^c$ (where $c$ is a constant) achieved in Ref. [95] to logarithmic scaling $\log(N)$. Beyond the studies on the QPE problem for quantum many-body systems, Du et al. [96, 97] developed efficient classical learners for estimating the linear properties of parameterized quantum circuits under specific conditions.

**DL algorithms for QPE.** Deep learning algorithms provide enhanced capability for recognizing complex patterns in classical data collected from quantum many-body systems, enabling predictions of complex properties such as entanglement entropy with improved accuracy. Current DL algorithms can be categorized into three primary learning paradigms: supervised learning (SL), semi-supervised learning (SSL), and self-supervised learning with fine-tuning (SSL-FT).

The supervised learning paradigm explores various neural architectures for effectively constructing classical representations of the quantum states, including restricted Boltzmann machines [26, 28, 98], multi-layer perceptions [40, 41, 48], convolutional neural networks [29, 43, 47], and attention-based neural networks [35, 42, 49, 50]. The parameterized neural network for state restriction is optimized to approximate the target values of quantum properties in the training dataset under specific loss functions, such as mean square loss or cross-entropy loss.

The SSL and SSL-FT paradigms address the challenge of limited labeled data, allowing training with datasets comprising both labeled and unlabeled data. Tang et al. proposed a teacher-student model for semi-supervised training [51] and a large language model-style quantum task-agnostic pretraining and fine-tuning paradigm [52]. Both paradigms incorporate SL approaches for the labeled dataset, but differ in their sequence: SSL applies supervised learning before training with unlabeled data, whereas SSL-FT employs supervised fine-tuning after the self-supervised pretraining stage, which will be detailed in Appendix C.

## C    Implementation details of various learning paradigms for QPE

In this section, we present the implementation details of the three learning paradigms employed in the experiments: supervised learning (SL), semi-supervised learning (SSL), and self-supervised learning with fine-tuning (SSL-FT). All paradigms share a common neural network architecture comprising three main components: the input (encoding) layer, the hidden layer, and the output layer. To ensure a fair comparison of model performance, we adopt an identical architecture for the input and hidden layers across all learning paradigms as introduced in Appendix C.1, while the output layer is customized to suit the specific requirements of each paradigm as elucidated in Appendix C.2. Finally, we elucidate the adjustment strategy of the threshold $\tau$ in the model evaluation step of AiDE-Q in Appendix C.3.

### C.1    Input layer and hidden layer

We follow the attention-based neural architecture proposed in Ref. [52] to design the input and hidden layers for representing quantum states.

**Input layer.** To capture the hidden patterns of the quantum system, the input layer incorporates three types of embeddings: token embeddings, condition embeddings, and position embeddings.

Each measurement outcome $o_j \in \{0, 1\}$ under the corresponding Pauli measurement $M_j \in \{X, Y, Z\}$ on the $j$-th qubit yields six possible combinations: $(X, 0)$, $(X, 1)$, $(Y, 0)$, $(Y, 1)$, $(Z, 0)$, and $(Z, 1)$. These pairs can be bijectively mapped to integers $\sigma \in [6]$. Consequently, the measurement outcomes $(M, o)$ can be represented as a tokenized measurement string $\sigma(M, o)$, where each element $\sigma(M_i, o_i) \in [6]$ resembles a token in natural language processing (NLP). The token embedding layer applies a linear transformation to the measurement string, augmented with a start token $s$, producing a feature tensor $E_t \in \mathbb{R}^{B_t \times (N+1) \times d}$, where $B_t$ denotes the batch size and $d$ is the feature dimension. A feed-forward network (FFN) with one hidden layer is then used to embed the physical condition $p$ into a vector $E_c \in \mathbb{R}^{B_t \times d}$. The final input embedding is computed as the element-wise sum

$$E_{\text{out}} = E_t + E_c + E_p,$$

where $E_p$ denotes the position embeddings, as described in Ref. [99]. The resulting tensor $E_{\text{out}}$ is passed to the hidden layers for further processing.

**Hidden Layer.** The hidden layer consists of a multi-layer Transformer decoder, following the architecture of Ref. [99]. It takes $E_{\text{out}}$ as input and produces output $F \in \mathbb{R}^{B_t \times (N+1) \times d}$, which encodes high-level representations of both the measurement strings and the conditional physical parameters. For additional architectural details, refer to Ref. [52].

## C.2   Output Layer and loss function

We now separately describe the neural architectures of the output layer and the associated loss functions under the explored three learning paradigms. Specifically, the design of the output layer is tailored to accommodate the specific loss function used in each paradigm.

**SL paradigms.** Recall that in the SL paradigm, the training dataset for a batch is given by $\mathcal{S}_{\text{SL}} = \{(x^{(i)}, y^{(i)})\}_{i=1}^{B_t}$, where each input $x^{(i)} = (p^{(i)}, M^{(i)}, o^{(i)})$ is derived from quantum states $\rho(p^{(i)})$ with different parameters $p^{(i)}$, and $y^{(i)} \in \mathbb{R}^K$ represents the target label vector of dimension $K$. The loss function for a training batch is defined as

$$\mathcal{L}_{\text{SL}} = \frac{1}{B_t} \sum_{i=1}^{B_t} \left\| f_{\text{SL}}(x^{(i)}) - y^{(i)} \right\|^2. \tag{16}$$

To support this loss function, the output of the neural network for each input $x^{(i)}$ must be a $K$-dimensional vector. Accordingly, the output layer consists of a feature aggregation module followed by a linear projection. Specifically, hidden features $F \in \mathbb{R}^{B_t \times (N+1) \times d}$ are aggregated along the second axis to produce a feature representation $F' \in \mathbb{R}^{B_t \times d}$ for each training example. This is followed by a linear projection into $F'' \in \mathbb{R}^{B_t \times K}$, with an optional task-specific activation function. For instance, a tanh activation is used when predicting correlation functions.

**SSL paradigms.** In the SSL setting, the training dataset includes both labeled and unlabeled data, denoted by $\mathcal{S}_{\text{SSL}} = \{(x^{(i)}, y^{(i)})\}_{i=1}^{n_l} \cup \{x^{(i)}\}_{i=n_l+1}^{n}$, where $n_l$ is the number of labeled samples. A teacher-student framework [71] is adopted, consisting of a student model $f_{\text{SSL}}$ and a teacher model $f'_{\text{SSL}}$, both sharing the same architecture. The loss function for the student model in a given batch yields

$$\mathcal{L}_{\text{SSL}} = \frac{1}{B_l} \sum_{i=1}^{B_l} \left\| f_{\text{SSL}}(x^{(i)}) - y^{(i)} \right\|^2 + \frac{\lambda}{B_t - B_l} \sum_{i=B_l+1}^{B_t} \left\| f_{\text{SSL}}(x^{(i)}) - f'_{\text{SSL}}(x^{(i)}) \right\|^2, \tag{17}$$

where $\lambda$ refers to the consistency weight, $B_l$ is the number of labeled samples in the batch, and $B_t$ is the total batch size. The proportion of labeled samples per batch is kept consistent with their proportion in the overall training dataset, i.e., $B_l/B_t = n_l/n$. The teacher model's parameters are updated as an exponential moving average of the student model's parameters, following the approach in Ref. [71].

Since the outputs of the DL models in the SSL paradigm play the same role of $K$-dimensional predicted labels as that in the SL paradigm, the architectures of the teacher and student networks are designed identically to the ones used in the SL setting.

**SSL-FT paradigms.** The SSL-FT paradigm adopts a two-stage training procedure. In the first stage, a self-supervised pretraining is performed using the unlabeled dataset $\{x = (p, M, o)\}$, where the

goal is to model the classical probability distribution $\mathbb{P}(\boldsymbol{\sigma}(\boldsymbol{M}_1, \boldsymbol{o}_1), \cdots, \boldsymbol{\sigma}(\boldsymbol{M}_N, \boldsymbol{o}_N))$ associated with a quantum state $\rho(\boldsymbol{p})$ with fixed parameters $\boldsymbol{p}$. This is achieved by minimizing the average negative log-likelihood loss

$$\mathcal{L}_{\mathrm{SF}} = \frac{1}{B_t - B_l} \sum_{i=1}^{B_t - B_l} - \log \mathbb{P}\left(\boldsymbol{\sigma}(\boldsymbol{M}_1^{(i)}, \boldsymbol{o}_1^{(i)}), \cdots, \boldsymbol{\sigma}(\boldsymbol{M}_N^{(i)}, \boldsymbol{o}_N^{(i)}) \mid \boldsymbol{p}^{(i)}\right), \tag{18}$$

which corresponds to maximizing the conditional likelihood of observed measurement outcomes.

To produce valid probability distributions, the output layer consists of a linear transformation followed by a softmax activation, ensuring normalization

$$\sum_{(\boldsymbol{M}_1, \boldsymbol{o}_1)} \cdots \sum_{(\boldsymbol{M}_N, \boldsymbol{o}_N)} \mathbb{P}(\boldsymbol{\sigma}(\boldsymbol{M}_1, \boldsymbol{o}_1), \cdots, \boldsymbol{\sigma}(\boldsymbol{M}_N, \boldsymbol{o}_N)) = 1.$$

In the second stage, the pretrained model is fine-tuned on the labeled dataset $\{(\boldsymbol{x}, \boldsymbol{y})\}$ using the same supervised loss and output layer design as in the SL paradigm, namely

$$\mathcal{L}_{\mathrm{FT}} = \frac{1}{B_t} \sum_{i=1}^{B_t} \left\| f_{\mathrm{SF}}(\boldsymbol{x}^{(i)}) - \boldsymbol{y}^{(i)} \right\|^2, \tag{19}$$

where the initial parameters of the input and hidden layers of $f_{\mathrm{SF}}$ inherit the optimized parameters during the pretraining stage.

### C.3 Adjustment strategy of the threshold $\tau$ in the model evaluation step of AiDE-Q

We first recall the step of model evaulation in AiDE-Q. As introduced in the maintext, given the hybrid dataset $\mathcal{S}_{\mathrm{hyd}}$, AiDE-Q iteratively updates the learning model $f_{\boldsymbol{\theta}(t)}$ and a high-quality dataset $\mathcal{S}_h(t) \subset \mathcal{S}_{\mathrm{hyd}}$. Specifically, training examples whose variance $\mathrm{Var}(\tilde{\boldsymbol{y}})$ is below a given threshold $\tau = \tau_0$ are identified as high-quality data and added to the high-quality dataset $\mathcal{S}_h(t+1)$.

The explanation of the strategy for adjusting the threshold $\tau$ is as follows. If the performance of $f_{\boldsymbol{\theta}(t+1)}$ decreases compared to $f_{\boldsymbol{\theta}(t)}$, AiDE-Q raises the threshold $\tau$ in the consistency-check module, re-initiating the high-quality labeled data collection, model training, and evaluation process until the model's performance on the validation dataset $\mathcal{S}_{\mathrm{val}}$ improves. In the main text, the threshold $\tau$ is raised to update the dataset $\mathcal{S}_h(t+1)$ by removing half of the newly added high-quality data in $\mathcal{S}_h(t+1) \backslash \mathcal{S}_h(t)$ that exhibit low consistency levels. Once the performance of $f_{\boldsymbol{\theta}(t+1)}$ improves, the threshold $\tau$ is reset to the initial value $\tau_0$ for the following high-quality data collection.

**Remark.** We remark that the threshold adjustment strategy in AiDE-Q is flexible. Alternative strategies, beyond those presented above, can be integrated into AiDE-Q.

## D Integration of AiDE-Q with machine learning models

In this section, we briefly review the machine learning (ML) models for QPE [36, 95], and relate them to the supervised learning (ML) paradigm. In this regard, the integration of ML models with AIDE-Q could follow the same manner as in the SL paradigm introduced in the main text.

We follow the conventions of Ref. [95] to introduce the ML model for QPE. In particular, the ML model considers the training dataset $\{(\boldsymbol{p}^{(i)}, \hat{\rho}(\boldsymbol{p}^{(i)}))\}_{i=1}^n$, where $\boldsymbol{p}^{(i)}$ are the sampled physical parameters, and $\hat{\rho}(\boldsymbol{p}^{(i)})$ refer to the classical shadow constructed with $(\boldsymbol{M}, \boldsymbol{o})$, as defined in Eq. 13. The classical ML models are trained on the size-$n$ training data, such that when given the input $\boldsymbol{p}^{(i)}$, the ML model can produce a classical representation $h(\boldsymbol{p}^{(i)})$ that approximates $\hat{\rho}(\boldsymbol{p}^{(i)})$. During prediction, the classical ML produces $h(\boldsymbol{p})$ for new values of $\boldsymbol{p}$ different from those in the training data. In particular, the predicted output of the trained classical ML models can be written as the extrapolation of the training data using a learned kernel $\kappa(\boldsymbol{p}, \boldsymbol{p}^{(i)}) \in \mathbb{R}$,

$$h(\boldsymbol{p}) = \frac{1}{n} \sum_{i=1}^n \kappa(\boldsymbol{p}, \boldsymbol{p}^{(i)}) \hat{\rho}(\boldsymbol{p}^{(i)}). \tag{20}$$

The ground state properties are then estimated using these predicted classical representations $h(\boldsymbol{p})$. Specifically, $f_{\mathrm{ML}}(\rho) = \mathrm{tr}(Oh(\boldsymbol{p}))$ can be predicted efficiently whenever $O$ is a sum of few-body

operators. In this regard, the trained classical ML for specific properties $\hat{\boldsymbol{y}}^{(i)} = \text{tr}(O\hat{\rho}(\boldsymbol{p}^{(i)}))$ could be written as

$$f_{\text{ML}}(\boldsymbol{p}) = \frac{1}{n} \sum_{i=1}^{n} \kappa(\boldsymbol{p}, \boldsymbol{p}^{(i)}) \hat{\boldsymbol{y}}^{(i)}. \tag{21}$$

Notably, the kernel function could be represented as the inner product of two feature vectors $\Phi(\boldsymbol{p})$ and $\Phi(\boldsymbol{p}^{(i)})$ according to Mercer's theorem [100]. In this regard, the learning model $f_{\text{ML}}(\boldsymbol{p})$ relates to the optimal solution of the following supervised learning problem

$$f_{\text{ML}}(\boldsymbol{p}) = \underset{f(\boldsymbol{p})=\boldsymbol{\omega}\cdot\Phi(\boldsymbol{p})}{\arg\min} \mathcal{L}_{\text{ML}}(\boldsymbol{\omega}) := \underset{f(\boldsymbol{p})=\boldsymbol{\omega}\cdot\Phi(\boldsymbol{p})}{\arg\min} \frac{1}{n} \sum_{i=1}^{n} \left( f(\boldsymbol{p}^{(i)}) - \hat{\boldsymbol{y}}^{(i)} \right)^2, \tag{22}$$

where $\boldsymbol{\omega}$ refers to the optimized parameters. For the neural tangent kernel $\kappa(\boldsymbol{p}, \boldsymbol{p}^{(i)})$ [37], the feature vector $\Phi(\boldsymbol{p})$ corresponds to the output of a deep neural network with large hidden layers [101].

To summarize, the ML models used in QPE naturally fit within the supervised learning paradigm described in Appendix C. Consequently, integrating these ML models with AIDE-Q can proceed in the same manner as outlined for the SL paradigm in the main text.

## E  Convergence analysis of AiDE-Q

In this section, we try to provide a theoretical understanding of the convergence of AiDE-Q by employing the theoretical results established for self-training algorithms in a special case [102]. In the following, we first briefly recap the results from Ref. [102], and then introduce the convergence of AiDE-Q based on these results.

**Convergence performance of self-training algorithms.** The self-training algorithm follows a similar iterative training manner as AiDE-Q on a hybrid dataset $\mathcal{S}_{\text{hyd}} = \mathcal{S}_L \cup \mathcal{S}_U$ by generating pseudo label of unlabeled data $\tilde{\boldsymbol{x}} \in \mathcal{S}_U$, while without considering the identification of high-quality data. Let the inputs of both the labeled and unlabeled data follow the normal distribution, $\boldsymbol{x} \in \mathcal{S}_L \sim \text{N}(0, \delta^2 \mathbb{I}_d)$ and $\tilde{\boldsymbol{x}} \in \mathcal{S}_U \sim \text{N}(0, \tilde{\delta}^2 \mathbb{I}_d)$. Let $\boldsymbol{W} \in \mathbb{R}^{d \times K}$ be the trainable weights of a one-hidden-layer neural network, and assume the existence of a ground-truth model with weights $\boldsymbol{W}^*$. Let $\boldsymbol{W}^{(0)}$ be model trained on the labeled data $\mathcal{S}_L$ only, and $\boldsymbol{W}^{(t)}$ be the model further trained on the unlabeled data $\mathcal{S}_U$ after $t$ iterations. Ref. [102] established the convergence performance of self-training algorithms by deriving an upper bound of the norm of differences between $\boldsymbol{W}^{(t)}$ and $\boldsymbol{W}^*$, i.e.,

$$\|\boldsymbol{W}^{(t)} - \boldsymbol{W}^*\|_{\text{F}} \leq [(1 + \frac{c_1}{\sqrt{n_l}} + \frac{c_2}{\sqrt{n_u}}) \cdot c_3(1 - \hat{\lambda})]^t \cdot \|\boldsymbol{W}^{(0)} - \boldsymbol{W}^*\|_{\text{F}},$$

where $n_l, n_u$ refer to the number of labeled and unlabeled data, $c_1, c_2, c_3$ could be regarded as constants, $\hat{\lambda} = \delta/(\delta + \tilde{\delta})$. It is observed that the prediction performance of $\boldsymbol{W}^{(t)}$ achieves a linear convergence regarding the iteration $t$, when $\hat{\lambda} > 1 - 1/(c_3)$. The results established for self-training algorithms also apply to AiDE-Q, as AiDE-Q follows the same iterative training manner with the only difference of using different unlabeled data during the iteration of $\boldsymbol{W}^{(t)}$.

**Convergence analysis of AiDE-Q.** We now understand these convergence results in the context of using AiDE-Q for QPE with a limited number of measurements. As discussed above, the model $W^{(t)}$ achieves linear convergence for $\hat{\lambda} > 1 - 1/(c_3\sqrt{K})$, otherwise the model performance will decrease as training on the unlabeled data proceeds. Moreover, the value $\hat{\lambda}$ is determined by the variances $\delta$ and $\tilde{\delta}$ for the labeled data $\boldsymbol{x}$ and unlabeled data $\tilde{\boldsymbol{x}}$. In this regard, the convergence of $\boldsymbol{W}^{(t)}$ of AiDE-Q is determined by the variance $\tilde{\delta}$ of the identified high-quality unlabeled data. For the problem of QPE, the variance of the input $\tilde{\boldsymbol{x}} = (\boldsymbol{p}, \tilde{\boldsymbol{o}})$ varies with the different physical parameters $p$ and the number of measurements $m_u$ for obtaining $\tilde{\boldsymbol{o}}$. In particular, a small number of measurements $m_u$ could lead to a large variance $\tilde{\delta}$ for some specific parameters $\{\boldsymbol{p}\}$ such that $\hat{\lambda} < 1 - 1/(c_3\sqrt{K})$, and hence decrease the model's prediction performance. The theoretical results provide a rationale for using AiDE-Q to improve the performance of deep learning models by identifying high-quality unlabeled data with low variance, as the variance of the input $\boldsymbol{x}$ is generally unknown.

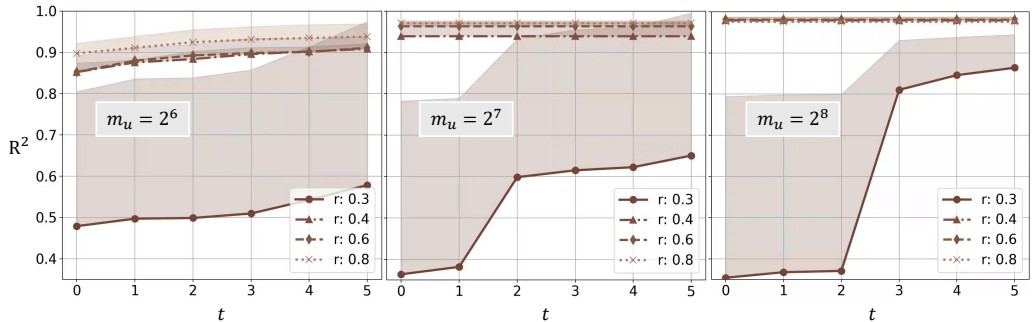

Figure 5: $R^2$ in entanglement entropy prediction for the 50-qubit Heisenberg XXZ model without using the physical parameters for constructing the training dataset. The panels from left to right correspond to the prediction performance for the number of measurements $m_u \in \{2^6, 2^7, 2^8\}$.

## F  Experimental setting and more experimental results

In this section, we conduct extensive experiments to evaluate the effectiveness of AiDE-Q across various hyperparameter settings, as well as different physical and chemical models. Furthermore, we also assess its robustness in the noisy settings, where the measurement data are obtained from noisy quantum processors.

### F.1  Experimental setting

**Hardware platform.** All the generation of training datasets are implemented by PastaQ [103] and ITensors [104] in the Julia language and run on classical device with Intel(R) Xeon(R) Gold 6226R CPU @ 2.90GHz and 256 GB memory. All deep learning models in various learning paradigms are implemented by Pytorch [105] and are trained on a single NVIDIA GeForce RTX 3090 with 24G graphics memory.

**Hyperparameter setting.** The deep learning (DL) models use the attention-based neural architectures described in Appendix C. The embedding dimension is set to $d = 128$, the number of attention heads is set to 4, and the number of hidden layers is set to $L = 2$. We optimize the DL model using the ADAM optimizer with a learning rate of $l = 10^{-3}$ and a batch size of $B_t = 64$. The maximum number of training epochs is set to 300. To mitigate overfitting, we employ an early stopping strategy. During the initial training stage, early stopping is applied after 100 epochs. In the subsequent training stage, with the updated high-quality dataset, early stopping is initiated after 30 epochs. Each configuration is run 5 times to report the average prediction performance.

### F.2  Experimental results for various hyperparameter settings in AiDE-Q

Here, we conduct extensive experiments to examine the effect of the hyperparameters in AiDE-Q, namely *the number of subsets for the consistency check s, the size of these subsets $|\mathcal{I}|$, and the variance threshold for data selection $\tau$*, where we drop the subscript of $\mathcal{I}_k$ for clarity. In particular, we focus on the 10-qubit Heisenberg XXZ model.

**Hyperparameter setting.** We set the number of subset as $s \in \{3, 5, 7\}$, the size of each subset as $|\mathcal{I}| = \alpha_{\mathcal{I}} m_u$ with $\alpha_{\mathcal{I}} \in \{1/8, 1/4, 1/2\}$ and $m_u$ being the number of measurements for the low-quality dataset $\mathcal{S}_U$, and the threshold $\tau \in \{95\%, 90\%, 80\%\}$ with $\beta\%$ referring to taking the threshold as consistency level at the top $1 - \beta\%$. The hyperparameters for constructing the hybrid dataset $\mathcal{S}_{\text{hyd}} = \mathcal{S}_L \cup \mathcal{S}_U$ are set to the same in our manuscript. In particular, the total number of training data is set as $n = 720$, the ratio of the number of high-quality data $n_l$ to the total dataset size $n$ is defined as $r := n_l/n = 0.4$. The number of measurements for the initial high-quality dataset $\mathcal{S}_L$ is set to $m_l = 2^{10}$, while the number of measurements for the initial low-quality dataset $\mathcal{S}_U$ is $m_u = 2^6$.

**Experimental results.** Table 3 presents $R^2$ values in predicting the entanglement entropy $S_A$ for the supervised learning (SL) models with and without incorporating AiDE-Q. It is observed that

Table 3: $R^2$ in predicting the entanglement entropy $S_A$ with various hyperparameters $(s, \alpha_{\mathcal{I}, \tau})$, where $A = [j]$ and $j \in [N-1]$. The initial ratio of high-quality data is set as $r = 0.4$. The best results are emphasized in blue while the second-best results are distinguished in orange.

| Method | s=5 | | | | | | | | |
|---|---|---|---|---|---|---|---|---|---|
| | $\alpha_{\mathcal{I}} = 1/8$ | | | $\alpha_{\mathcal{I}} = 1/4$ | | | $\alpha_{\mathcal{I}} = 1/2$ | | |
| | $\tau = 95\%$ | $\tau = 90\%$ | $\tau = 80\%$ | $\tau = 95\%$ | $\tau = 90\%$ | $\tau = 80\%$ | $\tau = 95\%$ | $\tau = 90\%$ | $\tau = 80\%$ |
| SL | 0.745 | 0.75 | 0.722 | 0.717 | 0.754 | 0.741 | 0.726 | 0.721 | 0.788 |
| SL w. DE | 0.834 | 0.801 | 0.805 | 0.853 | 0.854 | 0.809 | 0.82 | 0.815 | 0.857 |
| Improvement | 0.089 | 0.051 | 0.083 | 0.136 | 0.1 | 0.068 | 0.093 | 0.094 | 0.07 |

| Method | $\tau = 90\%$ | | | | | | | | |
|---|---|---|---|---|---|---|---|---|---|
| | $s = 3$ | | | $s = 5$ | | | $s = 7$ | | |
| | $\alpha_{\mathcal{I}} = 1/8$ | $\alpha_{\mathcal{I}} = 1/4$ | $\alpha_{\mathcal{I}} = 1/2$ | $\alpha_{\mathcal{I}} = 1/8$ | $\alpha_{\mathcal{I}} = 1/4$ | $\alpha_{\mathcal{I}} = 1/2$ | $\alpha_{\mathcal{I}} = 1/8$ | $\alpha_{\mathcal{I}} = 1/4$ | $\alpha_{\mathcal{I}} = 1/2$ |
| SL | 0.723 | 0.722 | 0.779 | 0.75 | 0.754 | 0.721 | 0.774 | 0.73 | 0.739 |
| SL w. DE | 0.806 | 0.834 | 0.829 | 0.801 | 0.854 | 0.815 | 0.849 | 0.825 | 0.807 |
| Improvement | 0.083 | 0.112 | 0.05 | 0.051 | 0.1 | 0.094 | 0.075 | 0.095 | 0.068 |

| Method | $\alpha_{\mathcal{I}} = 1/4$ | | | | | | | | |
|---|---|---|---|---|---|---|---|---|---|
| | $\tau = 95\%$ | | | $\tau = 90\%$ | | | $\tau = 80\%$ | | |
| | $s = 3$ | $s = 5$ | $s = 7$ | $s = 3$ | $s = 5$ | $s = 7$ | $s = 3$ | $s = 5$ | $s = 7$ |
| SL | 0.753 | 0.717 | 0.78 | 0.722 | 0.754 | 0.73 | 0.786 | 0.741 | 0.71 |
| SL w. DE | 0.851 | 0.853 | 0.836 | 0.834 | 0.854 | 0.825 | 0.822 | 0.809 | 0.785 |
| Improvement | 0.098 | 0.136 | 0.056 | 0.112 | 0.1 | 0.095 | 0.036 | 0.068 | 0.075 |

the $R^2$ of SL models is improved with incorporating AiDE-Q in all hyperparameter settings up to 0.136 at $(s, |\mathcal{I}|, \tau) = (5, m_u/4, 0.05)$. We further separately explore the effect of varying the three hyperparameters $s, |\mathcal{I}|, \tau$ on the improvement of incorporating AiDE-Q into SL models. In particular, the experimental results show that for a fixed number of subsets $s \in \{3, 5\}$, Moreover, for the fixed threshold $\tau = 90\%$, the maximal improvement of $R^2$ is always achieved at $|\mathcal{I}| = m_u/4$ for all different setting of $s \in \{3, 5, 7\}$. Additionally, for a fixed size of subset $|\mathcal{I}| = m_u/4$ and a large threshold $\tau \in \{90\%, 95\%\}$, setting the number of subset $s = 5$ achieves the maximal improvement of $R^2$. These results suggest that to maximize the power of AiDE-Q, the size of subsets $|\mathcal{I}|$ and the number of subsets $s$ should be set moderately to improve the effectiveness of the consistency-check method for identifying high-quality data labels.

### F.3 Experimental results for various data types

In this section, we examine the prediction performance of the deep learning (DL) model trained on a dataset that does not include physical parameters as input. Specifically, instead of using the training dataset with inputs $\boldsymbol{x}(\boldsymbol{p}^{(i)}) = (\boldsymbol{p}^{(i)}, \boldsymbol{o}^{(i)}, \boldsymbol{M}^{(i)})$ as defined in Eq.(2), we construct the dataset with inputs $\boldsymbol{x}^{(i)} = (\boldsymbol{o}^{(i)}, \boldsymbol{M}^{(i)})$, where the physical parameters $\boldsymbol{p}^{(i)}$ are assumed to be unknown during the training process. The experimental results for predicting the entanglement entropy of the 50-qubit Heisenberg XXZ model are presented in Fig. 5. It is observed that the prediction performance can be consistently improved by AiDE-Q when the dataset contains a small ratio of high-quality data. For instance, with a large number of measurements ($m_u = 2^8$) or low-quality data and a small ratio of high-quality data ($r = 0.3$), the coefficient of determination $R^2$ improves from approximately 0.36 to 0.88. These experimental results demonstrate the effectiveness of AiDE-Q for handling various types of training data.

### F.4 Experimental results for noisy quantum states

In this section, we consider that the measurement data are collected from noisy quantum states. In the following, we separately introduce the construction of the training dataset for noisy quantum states, the hyperparameter setting, and the experimental results. All contents in this reply have been updated to the revised manuscript.

**Training dataset construction from noisy quantum states.** We first formulate the form of the training dataset when considering the quantum system noise. In particular, we employ the depolarization noise model $\mathcal{N}_q$ to characterize quantum system noise in the worst-case scenario. The prepared

Table 4: $R^2$ in predicting the entanglement entropy $S_A$, where $A = [j]$ and $j \in [N-1]$. The initial ratio of high-quality data is set as $r = 0.4$. In each setting of $m_u \in \{2^6, 2^7\}$, the best results are emphasized in blue while the second-best results are distinguished in orange.

| Method | $m_u = 2^6$ | | | | $m_u = 2^7$ | | | |
|---|---|---|---|---|---|---|---|---|
| | $q = 0$ | $q = 0.1$ | $q = 0.3$ | $q = 0.5$ | $q = 0$ | $q = 0.1$ | $q = 0.3$ | $q = 0.5$ |
| SL | 0.754 | 0.783 | 0.701 | 0.402 | 0.734 | 0.725 | 0.74 | 0.462 |
| SL w. DE | 0.854 | 0.863 | 0.769 | 0.457 | 0.82 | 0.789 | 0.808 | 0.542 |
| Improvement | 0.10 | 0.086 | 0.064 | 0.035 | 0.085 | 0.062 | 0.068 | 0.08 |

Table 5: $R^2$ in predicting the entanglement entropy $S_A$, two-point correlations $\mathcal{C}_{1j}^x$ and $\mathcal{C}_{1j}^z$ of 9-qubit cluster Ising model, where $A = [j]$ and $j \in [N-1]$. The number of measurements for low-quality data is set as $m_u = 2^6$. The best results are emphasized in blue while the second-best results are distinguished in orange.

| Method | $S_A$ | | | $\mathcal{C}_{1j}^x$ | | | $\mathcal{C}_{1j}^z$ | | |
|---|---|---|---|---|---|---|---|---|---|
| | $r = 0.4$ | $r = 0.6$ | $r = 0.8$ | $r = 0.4$ | $r = 0.6$ | $r = 0.8$ | $r = 0.4$ | $r = 0.6$ | $r = 0.8$ |
| SL | 0.226 | 0.356 | 0.51 | 0.902 | 0.928 | 0.938 | 0.903 | 0.925 | 0.958 |
| SSL4Q | 0.422 | 0.448 | 0.489 | 0.948 | 0.932 | 0.952 | 0.958 | 0.964 | 0.952 |
| LLM4QPE | 0.29 | 0.322 | 0.436 | 0.902 | 0.922 | 0.949 | 0.909 | 0.948 | 0.954 |
| NTK | - | - | - | 0.962 | 0.966 | 0.97 | 0.291 | 0.31 | 0.341 |
| CS | - | - | - | 0.829 | 0.829 | 0.829 | 0. | 0. | 0. |
| SL w. DE | 0.563 | 0.791 | 0.873 | 0.956 | 0.965 | 0.976 | 0.965 | 0.98 | 0.984 |

ground state under depolarization noise model $\mathcal{N}_q$ refers to

$$\tilde{\rho} = \mathcal{N}_q[\rho] = (1-q)\rho + q\frac{\mathbb{I}}{2^N},$$

where $0 \le q \le 1$ refers to the collected classical input about the noisy ground state $\tilde{\rho}(\boldsymbol{p}^{(i)}) = \mathcal{N}_q[\rho(\boldsymbol{p}^{(i)})]$ with physical parameters $\boldsymbol{p}^{(i)}$, and $\tilde{\boldsymbol{o}}^{(i)}$ is the measurement output of the Pauli operator $\boldsymbol{M}^{(i)}$. Here, $m_l, m_u$ refer to the number of measurements with $m_l < m_u$. Let $\tilde{\boldsymbol{y}}^{(i)}$ be the noisy label of $\tilde{\boldsymbol{x}}^{(i)}$ with the noise coming from the statistical measurement error and the quantum system noise.

**Hyperparameter setting.** The experiment setup for evaluating the performance of AiDE-Q is as follows. The hybrid dataset $\tilde{\mathcal{S}}_{\text{hyd}}$ is collected from noisy quantum states, focusing on the 10-qubit Heisenberg XXZ model. We set the noise rate of the depolarization noise model $\mathcal{N}_q$ as $q \in \{0.1, 0.3, 0.5\}$. The other hyperparameters are kept the same as those in our manuscript. In particular, the total number of training data is set as $n = 720$, the ratio of the number of high-quality data $n_l$ to the total dataset size $n$ is defined as $r := n_l/n = 0.4$. The number of measurements for the initial high-quality dataset $\mathcal{S}_L$ is set to $m_l = 2^{10}$, while for the initial low-quality dataset $\mathcal{S}_U$, the number of measurements is $m_u \in \{2^6, 2^7\}$.

**Experimental results.** Table 4 presents the $R^2$ values in predicting the entanglement entropy $S_A$ for the supervised learning models with and without incorporating AiDE-Q. It is observed that while the prediction performance of supervised learning models significantly decreases for a large noise rate $q = 0.5$, *AiDE-Q could still effectively improve the prediction performance of learning models for various noise levels* $q \in \{0.1, 0.3, 0.5\}$. The improvement in $R^2$ decreases from $0.1$ for the noiseless case of $q = 0$ to $0.035$ for the noisy case with $q = 0.5$, when the number of measurements is small for the low-quality data, namely $m_u = 2^6$. On the other hand, for a larger number of measurements $m_u = 2^7$, the AiDE-Q could remain a relatively large improvement of $R^2$, namely $0.08$, even when the noise rate reaches $q = 0.5$. This indicates that the improvement of prediction performance raised from AiDE-Q is robust to the quantum system noise for a large number of measurements $m_u$.

### F.5 More experimental results for the cluster-Ising model

In this section, we present additional experimental results for the 9-qubit cluster Ising model, following the same format as the Heisenberg XXZ model results discussed in the main text. The hyperparameter settings used to construct the training dataset are consistent with those described in the main text.

**Supervised learning models integrated with AiDE-Q outperform all reference models.** Table 5 presents the coefficient of determination $R^2$ for predicting entanglement entropy and two-point correlations in the 9-qubit cluster Ising model. The results show similar trends to those observed

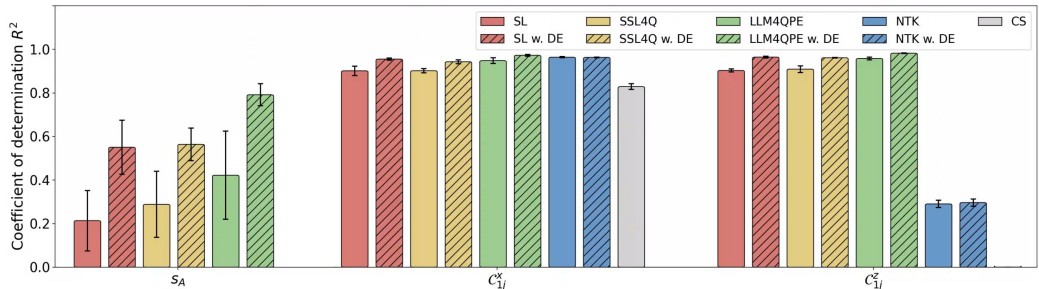

Figure 6: $R^2$ of reference models with and without integrating AiDE-Q in predicting entanglement entropy $S_A$, two-point correlations $\mathcal{C}^x_{1j}$ and $\mathcal{C}^z_{1j}$ of 9-qubit cluster Ising model, where $A = [j]$ and $j \in [N-1]$. The initial ratio of high-quality data and the number of measurements for low-quality data is set as $r = 0.4$ and $m_u = 2^6$.

Table 6: $R^2$ in predicting the entanglement entropy $S_A$ of $H_4$ molecular systems with $A = [j]$ and $j \in [N-1]$. The best results are emphasized in blue while the second-best results are distinguished in orange.

| Method | $m_u = 2^6$ | | | $m_u = 2^7$ | | | $m_u = 2^8$ | | |
|---|---|---|---|---|---|---|---|---|---|
| | $r = 0.4$ | $r = 0.6$ | $r = 0.8$ | $r = 0.4$ | $r = 0.6$ | $r = 0.8$ | $r = 0.4$ | $r = 0.6$ | $r = 0.8$ |
| SL | 0.085 | 0.178 | 0.178 | 0.196 | 0.254 | 0.275 | 0.127 | 0.167 | 0.225 |
| SSL4Q | 0.251 | 0.109 | 0.404 | 0.203 | 0.224 | 0.408 | 0.082 | 0.165 | 0.305 |
| LLM4QPE | 0.033 | 0.204 | 0.238 | 0.098 | 0.144 | 0.205 | 0.077 | 0.186 | 0.243 |
| SL w. DE | 0.405 | 0.619 | 0.679 | 0.478 | 0.638 | 0.734 | 0.5 | 0.632 | 0.751 |
| SSL4Q w. DE | 0.495 | 0.508 | 0.709 | 0.507 | 0.546 | 0.704 | 0.543 | 0.57 | 0.694 |
| LLM4QPE w. DE | 0.293 | 0.58 | 0.618 | 0.473 | 0.572 | 0.638 | 0.502 | 0.584 | 0.64 |

for the Heisenberg XXZ model, where the baseline SL model integrated with AiDE-Q achieves a significant performance improvement compared to the standalone SL model, outperforming all reference learning models across different properties and various ratio settings. Additionally, when the ratio of high-quality data is large $r = 0.8$, the baseline SL model even surpasses the advanced SSL4Q and LLM4QPE models in predicting entanglement entropy. This suggests that directly training DL models with low-quality data can harm prediction performance, even when using advanced learning models. In contrast, the AiDE-Q-enhanced SL model, trained with the identified high-quality synthetic labeled data, significantly improves the best $R^2$ value across the reference models from 0.51 to 0.873. These results emphasize the importance of constructing high-quality synthetic labeled data when training DL models for quantum property estimation.

**AiDE-Q enhances the prediction performance of various DL-based models.** Fig. 6 presents numerical results of predicting quantum properties when integrating AiDE-Q into various learning models. The ratio of high-quality data is set to $r = 0.4$, and the number of measurements for low-quality data is $m_u = 2^6$. The results show that all learning models exhibit an improvement in $R^2$ after incorporating AiDE-Q into their baseline models, particularly in predicting the complex non-linear properties of entanglement entropy.

### F.6 Experimental results for the chemical molecular model

We conduct numerical simulations for predicting the properties of chemical molecular models, focusing on the $H_4$ molecule with varied inter-atomic length.

**Dataset Generation.** The Hamiltonian of a molecular system under Jordan-Wigner transformation is given by

$$H(\boldsymbol{p}) = \sum_{i=1}^{N} \sum_{\alpha \in \{x,y,z\}} c_i^{\alpha}(\boldsymbol{p}) \sigma_i^{\alpha} + \sum_{i,j=1}^{N} \sum_{\alpha,\beta \in x,y,z} c_{i,j}^{(\alpha,\beta)}(\boldsymbol{p}) \sigma_i^{\alpha} \sigma_j^{\beta}, \tag{23}$$

where $\sigma_i^z$ ($\sigma_i^x$) is the Pauli-Z (Pauli-X) operator acting on the $i$-th qubit, $c_i^{\alpha}(\boldsymbol{p}), c_{i,j}^{(\alpha,\beta)}(\boldsymbol{p})$ refer to the Pauli coefficients dependent on the inter-atomic length $\boldsymbol{p}$. This Hamiltonian could be constructed with the Pennylane package [106]. For the $H_4$ molecule, this system is characterized by an 8-qubit

Hamiltonian $H(\boldsymbol{p})$. Here, the inter-atomic length $\boldsymbol{p}$ is sampled from the range $0.7\text{Å}$ to $1.3\text{Å}$. The number of sampled inter-atomic lengths is set as $n = 1280$. Other hyperparameters used to build the hybrid dataset in Eq. (5) are set to be the same as those used in quantum many-body systems.

**Experimental results.** Table 6 shows the coefficient of determination $R^2$ of entanglement entropy prediction for the $H_4$ molecule model. The results demonstrate that integrating AiDE-Q into various learning-based models significantly enhances their predictive performance across different ratios of high-quality data $r \in \{0.4, 0.6, 0.8\}$ and measurement counts $m_u \in \{2^6, 2^7, 2^8\}$. In particular, the mean $R^2$ values for the SL, SSL4Q, and LLM4QPE models are improved from $0.187, 0.239, 0.158$ to $0.604, 0.586, 0.544$, respectively, after integrating AiDE-Q. These results confirm the effectiveness of AiDE-Q in enhancing the prediction ability of DL models for a variety of quantum systems.

