# OpenReview forum: "AiDE-Q: Synthetic Labeled Datasets Can Enhance Learning Models for Quantum Property Estimation"
_NeurIPS.cc/2025/Conference — NeurIPS 2025 poster_

### Official Review · Reviewer_Kvb1 · 2025-06-27

**Clarity:** 3
**Significance:** 3
**Originality:** 3
**Rating:** 4
**Confidence:** 3

**Summary:**

The paper introduces AiDE-Q, a data framework engine designed for machine learning tasks in so-called quantum property estimation, particularly in the context of quantum many-body systems. It enables the generation of “high-quality” training data by assigning reliable synthetic labels even to noisy data derived from a limited number of quantum measurements. The core technical innovation is a consistency-check mechanism that assesses the confidence of synthetic labels by evaluating their stability across randomly masked subsets of measurement data. The method is compatible with various deep learning paradigms and the authors provide experimental results on Heisenberg XXZ and cluster Ising systems.

**Questions:**

-Can the authors provide an estimate of the computational cost of running AiDE-Q for different problem sizes? Also, how should the key parameters—like the number of iterations— be chosen in order to obtain a target accuracy?
-Related to this, how should one choose the threshold used in the consistency check? Does it have a strong impact on performance and running time?
-Why can we expect the algorithm will perform well at scale?
-Finally, can the authors compare AiDE-Q with existing methods for improving data quality, such as [1], [2], [3] and explain in what ways their method is better or different?

**Ethical Concerns:**

["NO or VERY MINOR ethics concerns only"]

**Final Justification:**

Upon discussion with the reviewers, I have changed my opinion for the necessity of further theoretical analyses. I was also encoureged by the additional experiments by the team. Overall my opinion of the paper has improved and I am raising the score to boarderline accept.
However, given the high bar of NeurIPS, I believe the paper would have been significantly strenghtened with any attempt for a theoretical justification of the results. In particular it is not easy to assess the scaling of the method beyond still rather small sizes, and this critically impacts the real world value of this method.
Consequently my grade is increased to boarderline accept.

**Limitations:**

yes

**Quality:**

3

**Strengths And Weaknesses:**

Strenghts:
-The work introduces a novel framework to construct a dataset of accurate labeled data from a dataset containing inaccurate samples. The algorithm ensures that the quality of the training set does not degrade, as each iteration includes a validation step where the performance of the model is evaluated, and newly generated synthetic data are only accepted if they do not reduce performance.

-It addresses a very important problem.

-The consistency-check used to assess whether a synthetic label is sufficiently accurate does not require additional measurements on the training data, meaning it introduces no measurement overhead.

-The authors validate their framework through experiments predicting physical quantities on systems such as the Heisenberg XXZ model and the one-dimensional cluster-Ising model.

-The paper is well organized and the exposition is clear.


Weaknesses:

-The main weakness of the paper is that it does not include any theoretical analysis of the proposed algorithm. In particular, the authors do not discuss the computational cost of running AiDE-Q—for example, is a fixed number of iterations expected to be sufficient in more general cases? How should the algorithm scale with system size or data volume? Moreover, the effect of choosing different threshold values in the consistency check on the performance of the algorithm is not explored.
Why can we expect the algorithm will perform well at scale?

-It is also not clear how the method is particularly suited for quantum property estimation problems. Specifically, why should the proposed method outperform existing techniques in the classical machine learning literature that also aim to enhance data quality? For example references [1,2,3] are not discussed or compared against in the paper.

---

> ### Author Rebuttal · Authors · 2025-07-30
>
> We thank Reviewer Kvb1 for the valuable comments. As many of the points raised under **Weaknesses** and **Questions** overlap, we have consolidated similar concerns under unified headings for clarity, indexed as **W1–W5** and **Q1–Q5**. All concerns have been carefully considered and are addressed in detail below. We hope our responses adequately clarify the issues and help convey the merits of our work.
>
> **[W1 & Q1]: A key weakness is the lack of theoretical analysis, particularly regarding the computational cost of AiDE-Q. Can the authors estimate its cost for varying problem sizes?**
>
> **A1:** To provide a comprehensive response, we organize our reply around three key aspects.
>
> $\bullet$ *Why does the submission lack theoretical analysis?* We respectfully note that in the context of *deep learning–based quantum property estimation*, it is conventional to validate proposed methods through extensive and systematic **empirical studies**, rather than through formal **theoretical guarantees** about computation cost. This practice is consistent with prior works, including Refs. [a–d].
>
> The primary reason for this convention is that *deep learning theory typically offers only loose bounds and limited explanatory power regarding the superior performance of deep neural networks* [Refs. [e-f]]. Consequently, following the practices established in the deep learning community, research on deep learning–based quantum property estimation (**DL-QPE**) has predominantly focused on empirical results. In this regard, in the following response, we mainly focus on providing **empirical evidence** to address the reviewer's concerns.
>
> $\bullet$ *Computational cost of AiDE-Q*. We would like to clarify that the computational cost considered in this field is primarily concerned with **the total number of queries to quantum computers**, given the limited availability of quantum resources compared to classical computing resources. In addition, as outlined in Lines 265-270 of our manuscript, the query cost is jointly determined by the size of the training dataset $n$ and the number of measurements $m$.
>
> To better address the reviewer's concern,  the table below summarizes the **execution time** of AiDE-Q for various system sizes. Notably, the longest running time observed for the setting with N=50 is 3.6 hours, which indicates the efficiency of AiDE-Q.
> | N | 10 | 20 | 30 | 40 | 50 |
> |---|----|----|----|----|----|
> | Time (h) | 0.8 | 0.9 | 1.8 | 2.4 | 3.6 |
>
> $\bullet$ *Theoretical analysis of AiDE-Q*. While providing a theoretical analysis is not central to this study, we nonetheless seek to address the reviewer’s concern by presenting a theoretical analysis of the **convergence of AiDE-Q**. We refer the reviewer to the response **A1** to Reviewer bVXi for details. Here we briefly summarize the achieved results.
>
> Assume the existence of a ground-truth model  $W^{\*}$. Let $n_l$ and $n_u$ be the number of labeled and unlabeled data, and $W(0)$ be the model trained on labeled data. The trained model $W(t)$ after $t$ iterations of AiDE-Q yields
> $$||W^{\*}-W(t)|| \le [(1+c_1/\sqrt{n_l} + c_2 /\sqrt{n_u}) \cdot c(1-\hat{\lambda})]^t ||W^{\*}-W(0)\|\|,$$
> where $c_1,c_2,c$ are constants, and $\hat{\lambda}$ is a decreasing function of the variance $\tilde{\delta}$ of unlabeled data.
>
> For QPE, the variance $\tilde{\delta}$ is often large due to a small number of measurements $m_u$ in the unlabeled data. Notably, a large variance of $\tilde{\delta}$ with $c(1-\hat{\lambda}) >1 $ could harm the prediction perfromance. By contrast, AiDE-Q can well suppress $\tilde{\delta}$,  where $c(1-\hat{\lambda}) <1$ leads to linear convergence. These results undermine the importance of using AiDE-Q to advance DL-QPE.
>
> **[W1 & Q2]: Is a fixed number of iterations sufficient in general? How should key parameters (such as iteration count) be selected to achieve a desired accuracy**
>
> **A2:**  To better address the reviewer’s concerns, we begin by briefly restating the main objective of our submission, followed by a detailed response to the two questions raised.
>
> $\bullet$ *Main objective of our work*. We respectfully clarify that the primary focus of our study is to validate how AiDE-Q improves diverse DL-QPE models under **limited quantum query budgets**, rather than to analyze the runtime or iteration cost associated with its implementation.
>
> $\bullet$  *Is a fixed number of iterations ... more general cases?* Our experimental results (**Table 4**) show that *AiDE-Q consistently improves the performance of nearly all DL-QPE models under the chosen hyperparameter settings*.  Moreover, the number of iterations can be **flexibly adjusted** to ensure reliable convergence in general cases, provided that the hyperparameters are applied consistently across different DL-QPE models.
>
> $\bullet$ *How should the key parameters ... target accuracy?* We respectfully note that our key objective is not to identify optimal parameters to enable DL-QPE models to reach a specific target accuracy. As mentioned earlier, we aim to evaluate the relative performance gains brought by AiDE-Q under consistent training settings and limited quantum query budgets. To ensure a fair comparison, the hyperparameters used, including the number of iterations, are kept consistent for different backbone models.
>
> **[W1 & Q4]: How should AiDE-Q scale with system size or data volume? Why can we expect it will perform well at scale?**
>
> **A3:** As discussed in Response **A1**, we would like to re-emphasize that the conventions for studying the scalability in DP-QPE are through **systematic experiments** with varied qubit count $N$ and training dataset sizes $n$, as exemplified by Refs. [a-d].
>
> In line with these conventions, the scalability with respect to $N$ **has been studied** in our original manuscript, as illustrated in Lines 330-335 with experimental results shown in Fig. 4(a). The table below summarizes the main results, demonstrating that AiDE-Q achieves significant performance improvement with **linearly increasing $n$ and model parameters.**, thereby **validating its scalability**.
>
> | N | 10 | 20 | 30 | 40 | 50 |
> |---|----|----|----|----|----|
> | n| 720 | 1280 | 1860 | 2360 | 2740 |
> | Improvement of $\mathrm{R}^2$ by AiDE-Q | 0.103 | 0.078 | 0.072 | 0.058 | 0.056 |
>
> **[W1 & Q2]: The effect of choosing different threshold values in the consistency check is not explored. Does it have a strong impact on performance and running time?**
>
> **A4:** To address the reviewer's concerns, we conduct additional experiments to show the effect of various hyperparameters of AiDE-Q on its performance, including the *the number of subsets for the consistency check $s$, the size of these subsets $|\mathcal{I}|$, and the variance threshold for data selection $\tau$*.
>
> The table below presents the $\mathrm{R}^2$ for varying $|\mathcal{I}|$ and $\tau$, and we refer the reviewer to Response **A2** for reviewer bVXi for a detailed introduction of the hyperparameter settings and experimental results.
>
> | $\|\mathcal{I}\|=α_I m_u$ | α_I=1/8 | | | α_I=1/4 | | | α_I=1/2 | | |
> |--------|---------|---------|---------|---------|---------|---------|---------|---------|---------|
> | Threshold| τ=95% | τ=90% | τ=80% | τ=95% | τ=90% | τ=80% | τ=95% | τ=90% | τ=80% |
> | Improvement by AiDE-Q| **0.089** | 0.051 | 0.083 | **0.136** | 0.1 | 0.068 | 0.093 | **0.094** | 0.07 |
>
> These results suggest that to maximize the power of AiDE-Q, the size of subsets $|\mathcal{I}|$ and the number of subsets $s$ should be set moderately, and the threshold should be set to be relatively large.
>
> **[W2 & Q5]: It is also not clear how the method is particularly suited for QPE problems. Why should AiDE-Q outperform existing techniques in the classical ML literature, such as references [1,2,3]? Can the authors compare AiDE-Q with them?**
>
> **A5:** Let us separately address the reviewer's concerns.
>
> $\bullet$ *The applicability of AiDE-Q to QPE.* To address the reviewer's concerns,  we begin by recalling that the proposed AiDE-Q is a **simple but effective** framework that iteratively enhances DL models. AiDE-Q's applicability to QPE stems from its **compatibility with existing DL models developed for QPE**, such as those in Refs [a-d]. Furthermore, AiDE-Q’s ability to enhance data quality is driven by the proposed consistency-check method, which is designed to identify high-quality data. The conducted experiments have **validated the effectiveness of AiDE-Q** in enhancing the prediction performance of various DL models for QPE.
>
> $\bullet$  *The comparison with other classical ML methods for improving data quality.* Before addressing this concern, we would like to emphasize that AiDE-Q is the **first framework** specifically proposed to improve data quality for the task of QPE. This unique contribution is also highlighted in Reviewer bVXi's comments, where it is noted: *While the concept of a `data engine' or self-training with pseudo-labels exists in the broader machine learning literature, **its application to QPE is novel.***
>
> Furthermore, since the reviewer **did not provide references [1-3]**, we are unable to clearly compare AiDE-Q with these mentioned methods. We would be happy to incorporate them and **conduct corresponding comparative experiments** in the discussion period, should the reviewer kindly provide them.
>
> Lastly, we would like to highlight that a significant advantage of AiDE-Q is its **simplicity**, which allows it to be highly compatible with most existing DL models, thereby enhancing their performance.
>
> [a] Phys. Rev. Lett. 130, 210601 (2023).
>
> [b] Nat Commun 15, 8796 (2024).
>
> [c] "Unsupervised pretraining of quantum property estimation and a benchmark" ICLR (2024).
>
> [d] "Semi-supervised learning of quantum data with application to quantum state classification" ICML (2024).
>
> [e] "Understanding deep learning requires rethinking generalization" ICLR (2017).
>
> [f] Proc. Natl. Acad. Sci. U.S.A. 116 (32) 15849-15854 (2019).

---

> > ### Comment · Reviewer_Kvb1 · 2025-08-04
> >
> > I genuinely appreciate the authors' explanation and the additional work they have undertaken. I understand their point that, in deep learning, it is often challenging to provide strong theoretical guarantees. While I still believe that a good paper should strive to include some theoretical insights, I recognize and value the authors’ effort in this direction, as well as the expanded set of experiments they have provided. For these reasons, I have decided to raise my evaluation.
> >
> > I am sorry if some references were lost in the earlier version, they are given below.
> >
> > [1] Yoon, Jinsung, Sercan Arik, and Tomas Pfister. "Data valuation using reinforcement learning." International Conference on Machine Learning. PMLR, 2020.
> >
> > [2] Koh, Pang Wei, and Percy Liang. "Understanding black-box predictions via influence functions." International conference on machine learning. PMLR, 2017.
> >
> > [3] Ghorbani, Amirata, and James Zou. "Data shapley: Equitable valuation of data for machine learning." International conference on machine learning. PMLR, 2019.

---

> > > ### Author Response · Authors · 2025-08-04
> > >
> > > Dear Reviewer Kvb1
> > >
> > > We sincerely appreciate your positive feedback and your decision to raise the evaluation score. Your valuable insights have greatly contributed to the improvement of our paper.
> > >
> > > We are also grateful for providing the missing reference concerning classical machine learning techniques for improving data quality. We will include these references in the revised manuscript, highlighting the distinctions between these methods and our approach, as well as exploring how they could be adapted to further advance quantum property estimation.

---

### Official Review · Reviewer_bVXi · 2025-07-02

**Clarity:** 3
**Significance:** 3
**Originality:** 3
**Rating:** 4
**Confidence:** 3

**Summary:**

This paper introduces AiDE-Q, a framework designed to improve the performance of deep learning (DL) models for quantum property estimation (QPE) when faced with practical limitations on data acquisition, such as a finite number of quantum measurements. The core challenge addressed is that real-world datasets for QPE are often "hybrid," containing a small amount of high-quality data (many measurements, accurate labels) and a large amount of low-quality data (few measurements, noisy labels).

**Questions:**

Section 3.2 mentions that if the validation performance decreases, the variance threshold is raised to be more selective. Could you elaborate on the mechanism for adjusting this threshold?

**Ethical Concerns:**

["NO or VERY MINOR ethics concerns only"]

**Final Justification:**

Keeping the score.

**Limitations:**

Addressed

**Quality:**

3

**Strengths And Weaknesses:**

**Strengths**

The paper addresses a critical and practical bottleneck in the application of DL to quantum many-body physics: the high cost and inherent noise of data acquisition from quantum systems. The problem of learning from hybrid-quality datasets is relevant for near-term quantum devices. The proposed AiDE-Q framework offers a practical solution to leverage large but noisy datasets effectively.

While the concept of a "data engine" or self-training with pseudo-labels exists in the broader machine learning literature, its application to QPE is novel. The specific implementation, particularly the "consistency-check" method to gauge the quality of synthetic labels based on measurement data variance, is an original and clever contribution tailored to the problem domain.

The paper is well-written, clearly structured, and easy to follow. The problem setup is well-motivated (Figure 1), and the AiDE-Q framework is explained lucidly with the help of a detailed diagram (Figure 2). The experimental settings and results are presented clearly, making the contributions accessible.

**Weaknesses**

The paper is primarily empirical. there is no theoretical analysis to formalize this or provide guarantees on convergence or performance improvement. For instance, under what conditions does the iterative process guarantee not to accumulate errors from incorrectly labeled data?

The framework introduces several new hyperparameters, such as the variance threshold for data selection (set to the top 10%), the number of subsets for the consistency check, and the size of these subsets. The paper uses fixed values for these and provides limited analysis of their sensitivity. A more detailed study on how these choices impact performance would strengthen the paper's claims of robustness.

The method of raising the variance threshold when validation performance drops is mentioned but not detailed. This seems like a crucial heuristic to prevent model degradation. A more principled or detailed explanation of this adaptive mechanism would be beneficial. Is it a fixed increment, or does it depend on the magnitude of the performance drop?

---

> ### Author Rebuttal · Authors · 2025-07-31
>
> We thank Reviewer bVXi for their affirmative comments and valuable feedback. For clarity, we index the points mentioned under **Weaknesses** as **W1-W4**, and **Questions** as **Q1**. In particular, we have addressed the points raised in **W3** and **Q1** together, as they are similar. All concerns have been carefully considered and are addressed in detail below.
>
> **[W1]: There is no theoretical analysis to guarantee convergence or performance improvement. Under what conditions does the iterative process guarantee not accumulate errors from incorrectly labeled data?**
>
> **A1:** We appreciate the reviewer’s concern regarding the lack of theoretical guarantees for the proposed method. We would like to respectfully point out that in the domain of deep learning–based quantum property estimation (**DL-QPE**), it is standard practice to validate new approaches through **extensive empirical studies** rather than formal theoretical analysis [Refs. a–d]. This is largely due to the fact that current deep learning theory often yields only loose bounds and offers limited explanatory power for the observed success of deep neural networks [e–f].
>
> Although theoretical analysis is *not the primary focus* of our study, we aim to best address the reviewer's concerns by bridging the gap between theory and model design. In the following, we provide a theoretical explanation of the convergence of AiDE-Q by using established results for self-training algorithms in a special case [g].
>
> $\bullet$ *Convergence performance of self-training algorithms.* Let us briefly recap the results of Ref. [g].  Assume the existence of a ground-truth model $W^{\*}$. Let $W(0)$ be a model trained on the labeled data only, and $W(t)$ be the model further trained on the unlabeled data after $t$ iterations. Ref. [g] established the convergence performance of $W(t)$ by deriving an upper bound of the norm of differences between $W(t)$ and $W^{\*}$, i.e.,
>  $$ ||W(t) - W^{\*}|| \le [(1+c _1/\sqrt{n _l}+c_2/\sqrt{n _u})\cdot c (1-\hat{\lambda})]^t \cdot ||W(0) - W^{\*} |\|, $$
> where $n_l,n_u$ are the number of labeled and unlabeled data, $c_1,c_2,c$ could be regarded as constants, $\hat{\lambda}$ is a decreasing function of the variance $\tilde{\delta}$ of unlabeled data.
>
> $\bullet$ *Convergence analysis of AiDE-Q.* As AiDE-Q shares a similar training manner with self-training algorithms, the established results in [g] also apply to AiDE-Q. For QPE, the variance $\tilde{\delta}$ is often large due to a small number of measurements $m_u$ in the unlabeled data, which could harm the performance with $c(1-\hat{\lambda}) >1$.
>
> By contrast, AiDE-Q can well suppress $\tilde{\delta}$ by identifying the high-quality data, where $c(1-\hat{\lambda}) <1$ leads to linear convergence. These results undermine the importance of using AiDE-Q to advance DL-QPE.
>
> $\bullet$ *Error accumulation from incorrectly labeled data.* To prevent the accumulated errors caused by incorrectly labeled data, a module has been designed to remove harmful unlabeled data by increasing the variance threshold for identifying high-quality data when the model’s performance drops.
>
> **[W2]: The paper uses fixed values for these and provides limited analysis of their sensitivity. Conduct a detailed study on how these choices impact performance**
>
> **A2:** We appreciate the reviewer’s suggestion and have *conducted additional experiments* to examine how varying hyperparameters affect the performance of AiDE-Q. Specifically, we investigate the impact of three key hyperparameters: *the number of subsets $s$,  the size of these subsets $|\mathcal{I}|$, and variance threshold $\tau$*. For clarity, we first describe the experimental setup, followed by a presentation of the corresponding results. All content below reflects updates made in the revised manuscript.
>
> $\bullet$ *Hyperparameter settings.*  We apply AiDE-Q to predict the entanglement entropy $S_A$ of the $10$-qubit Heisenberg XXZ models. We adopt the supervised learning (SL) model introduced in our manuscript as the baseline for comparison.
>
> The experiment setup for evaluating the performance of AiDE-Q is as follows. The number of subsets is set to $s=3,5,7$.  The size of each subset is set to $|\mathcal{I}|=\alpha_{\mathcal{I}} m _u$ with $\alpha _{\mathcal{I}} = 1/8, 1/4, 1/2$ and $m_u$ being the number of measurements for the low-quality dataset $\mathcal{S}_U$. The threshold $\tau =$ 95\%, 90\%, 80\% with $\beta$% referring to taking the threshold as a consistency level at the top $1-\beta$%. The other hyperparameters, including $n,r,m_u,m_l$, are kept the same as those in our manuscript, and as those introduced in the response **A1** for Reviewer nKML.
>
> $\bullet$ *Experimental results.* We present the results in the following three tables to separately explore the effect of varying hyperparameters $s, |\mathcal{I}|, \tau$. The presented results include $\mathrm{R}^2$ values obtained from the SL models with and without incorporating AiDE-Q, as well as the difference between them, which are highlighted by labels 'SL',  'SL w. DE', and 'Improvement', respectively.
>
> 1) The table below shows the results of varying $\tau$ and $|\mathcal{I}|$ with a fixed $s=5$. An observation is that the $\mathrm{R}^2$ of SL models is improved with incorporating AiDE-Q in all hyperparameter settings up to $0.136$ at $(s,|\mathcal{I}|,\tau)=(5,m_u/4,95$%$)$. Moreover, setting a large threshold $\tau$ leads to a large improvement of $\mathrm{R}^2$ in most cases.
>
> | Method | α_I=1/8 | | | α_I=1/4 | | | α_I=1/2 | | |
> |--------|---------|---------|---------|---------|---------|---------|---------|---------|---------|
> | | τ=95% | τ=90% | τ=80% | τ=95% | τ=90% | τ=80% | τ=95% | τ=90% | τ=80% |
> | SL | 0.745 | 0.75 | 0.722 | 0.717 | 0.754 | 0.741 | 0.727 | 0.721 | 0.788 |
> | SL w. DE | 0.834 | 0.801 | 0.805 | 0.853 | 0.854 | 0.809 | 0.82 | 0.815 | 0.857 |
> | Improvement | **0.089** | 0.051 | *0.083* | **0.136** | *0.1* | 0.068 | *0.093* | **0.094** | 0.07 |
>
> 2) The table below presents the results of varying $|\mathcal{I}|$ and $s$ with a fixed $\tau=$90%. The results show that the maximal improvement of $\mathrm{R}^2$ is always achieved at $|\mathcal{I}|=m_u/4$ for $s=3,5,7$.
> | Method | s=3 | | | s=5 | | | s=7 | | |
> |--------|-----|-----|-----|-----|-----|-----|-----|-----|-----|
> | | α_I=1/8 | α_I=1/4 | α_I=1/2 | α_I=1/8 | α_I=1/4 | α_I=1/2 | α_I=1/8 | α_I=1/4 | α_I=1/2 |
> | SL | 0.723 | 0.722 | 0.779 | 0.75 | 0.754 | 0.721 | 0.774 | 0.73 | 0.739 |
> | SL w. DE | 0.806 | 0.834 | 0.829 | 0.801 | 0.854 | 0.815 | 0.849 | 0.825 | 0.807 |
> | Improvement | *0.083* | **0.112** | 0.05 | 0.051 | **0.1** | *0.094* | *0.075* | **0.095** | 0.068 |
>
> 3) The table below presents the results of varying $\tau$ and $s$ with a fixed $|\mathcal{I}|=m_u/4$. Namely, for a large threshold $\tau=$ 90%, 95%, the maximal improvement of $\mathrm{R}^2$ is consistently achieved at a small $s=3,5$.
>
> | Method | τ=95% | | | τ=90% | | | τ=80% | | |
> |--------|-------|-------|-------|-------|-------|-------|-------|-------|-------|
> | | s=3 | s=5 | s=7 | s=3 | s=5 | s=7 | s=3 | s=5 | s=7 |
> | SL | 0.753 | 0.717 | 0.78 | 0.722 | 0.754 | 0.73 | 0.786 | 0.741 | 0.71 |
> | SL w. DE | 0.851 | 0.853 | 0.836 | 0.834 | 0.854 | 0.825 | 0.822 | 0.809 | 0.785 |
> | Improvement | *0.098* | **0.136** | 0.056 | **0.112** | *0.1* | 0.095 | 0.036 | *0.068* | **0.075** |
>
> The achieved results suggest that to maximize the power of AiDE-Q, the size of subsets $s$ and the number of subsets $|\mathcal{I}|$ should be set moderately, and the threshold $\tau$ should be set to be relatively large.
>
> **[W3 & Q1]: The method of raising the variance threshold when validation performance drops is mentioned but not detailed. Elaborate on the mechanism for adjusting this threshold.**
>
> **A3:** To better address the reviewer's concerns, let us first recap AiDE-Q and then provide a detailed explanation.
>
>  $\bullet$ *Overview of AiDE-Q.* As introduced in Sec. 3.2, given a hybrid dataset $\mathcal{S} _{hyd}$, AiDE-Q iteratively updates the learning model $f _{\theta(t)}$ and a high-quality dataset $\mathcal{S} _{h}(t)\subset \mathcal{S} _{hyd}$. Specifically, training examples whose $\text{Var} (\hat{y})$ is below a given threshold $\tau=\tau _0$ are identified as *high-quality data* and added to the high-quality dataset $\mathcal{S} _h(t+1)$.
>
> $\bullet$ *Explanation of the strategy for adjusting the threshold $\tau$.* If the performance of $f _{\theta(t+1)}$ decreases compared to $f _{\theta(t)}$, AiDE-Q raises the threshold $\tau$ in the *consistency-check module*, re-initiating the high-quality labeled data collection, model training, and evaluation process until the model’s performance on the validation dataset $\mathcal{S} _{val}$ improves.
>
> In the main text, the threshold $\tau$ is raised to update the dataset $\mathcal{S} _h(t+1)$ by **removing half of the newly added high-quality data** in $\mathcal{S}_h(t+1)  \backslash \mathcal{S} _h(t)$ that exhibit low consistency levels. Once the performance of $f _{\theta(t+1)}$ improves, the threshold $\tau$ is reset to the initial value $\tau_0$ for the following high-quality data collection.
>
> We remark that **the threshold adjustment strategy in AiDE-Q is flexible**. Alternative strategies, beyond those presented above, can be integrated into AiDE-Q. We have appended the discussion in the updated version.
>
> **[W4]: Is it a fixed increment, or does it depend on the magnitude of the performance drop?**
>
> **A4:** The increment of the variance threshold $\tau$ is **not fixed and does not depend on the magnitude of the performance drop**. As explained in the response to **A3**, the purpose of adjusting $\tau$ is to control the proportion of identified high-quality data based on their variances. Therefore, the increment of $\tau$ is **determined by the magnitude of the variances computed over the identified high-quality data**.
>
> [a-f] Refer to the response for Reviewer Kvb1
> [g] ArXiv:2201.08514. ICLR 2022

---

> > ### Comment · Reviewer_bVXi · 2025-08-06
> > **Thank you!**
> >
> > Thanks the authors for providing the detailed rebuttal. I will keep my recommendation for the paper.

---

> > > ### Author Response · Authors · 2025-08-06
> > >
> > > Dear Reviewer bVXi,
> > >
> > > Thank you for your kind words and your thorough review of our paper. We appreciate your acknowledgment of our detailed responses and are glad that our efforts help address your concerns and questions.

---

### Official Review · Reviewer_nKML · 2025-07-03

**Clarity:** 3
**Significance:** 3
**Originality:** 3
**Rating:** 5
**Confidence:** 4

**Summary:**

The paper presents AiDE-Q, a method for enhancing the datasets used to train deep learning (DL) models to estimate the ground state properties of quantum systems. Exactly calculating the ground state properties of a quantum system is exponentially expensive on classical hardware due to theoretical constraints, and practically challenging on quantum hardware due to hardware noise and the limited availability of quantum hardware. As a result, DL models for ground state property estimation are typically trained on hybrid datasets consisting of a mix of high-quality data with highly accurate labels and low-quality data with very approximate labels. The high ratio of low to high quality data naturally leads to poor DL model performance.

To address this issue, the paper proposes the integration of high-quality synthetic data into hybrid datasets. AiDE-Q iteratively improves DL models by generating synthetic labels through a trained model and employing a consistency-check method to assess the quality of these labels. Experimental results demonstrate that AiDE-Q achieves up to a 14.2% improvement in model performance when predicting entanglement entropy and correlation in the Heisenberg XXZ model and cluster Ising model with up to 50 qubits.

**Questions:**

What happens when you add in noise?

**Ethical Concerns:**

["NO or VERY MINOR ethics concerns only"]

**Final Justification:**

**Rating:** 5.

**Justification:** This paper is a technically sound paper with the potential to have a medium-to-high impact on multiple subfields of AI (hybrid quantum-classical machine learning, synthetic dataset generation). The paper does not merit a higher grade as it is unlikely to have a groundbreaking impact on any subfield of AI.

**Limitations:**

No analysis is done in the presence of noise.

**Quality:**

3

**Strengths And Weaknesses:**

This paper is overall a strong paper. Using quantum data to help train machine learning models is a popular idea. However, the effectiveness of these models will likely be limited even with the advent of fault-tolerant quantum computers. That's because access to quantum hardware will likely remain limited for the foreseeable future as building and operating quantum computers will continue to be very expensive. Having methods for improving the training sets of DL models for estimating the ground state properties will be critical.

The authors provide strong (given the length constraints of NeurIPS) evidence that AiDE-Q does improve model training. The authors ran comparative studies using a number of state-of-the-art DL approaches to ground state property estimation, showing improvement in most cases with AiDE-Q.

AiDE-Q is conceptually simple, which helps streamline the paper's presentation. The authors do a good job at contextualizing their work and surveying the subfield of DL-based approaches to ground state property estimation.

**Weaknesses:** It would be nice to see simulations done on noisy quantum computers. The noise models can be reflective of early fault-tolerant devices (<10^-3), Pauli stochastic errors, some very good qubits, etc. Otherwise it is hard to evaluate AiDE-Q's short-to-medium-term relevance to the community.

I didn't really spot many weaknesses outside of a typo or two. "Basic of quantum computing" should be "Basics of quantum computing."

---

> ### Author Rebuttal · Authors · 2025-07-30
>
> We deeply thank Reviewer nKML for their affirmative comments and constructive feedback on our work. Since the concerns raised in the **Weakness** and **Question** sections both focus on AiDE-Q's performance in noisy scenarios, we have merged them under the index **W1 & Q1**, while the minor weakness is indexed as **W2**. All concerns have been carefully considered and are addressed in detail below. We hope our responses adequately address your concerns.
>
> **[W1 & Q1]: It would be nice to see simulations done on noisy quantum computers. The noise models can be reflective of early fault-tolerant devices ($<10^{-3}$), Pauli stochastic errors, some very good qubits, etc. Otherwise it is hard to evaluate AiDE-Q's short-to-medium-term relevance to the community. What happens when you add in noise?**
>
> **A1**: We thank the reviewer for their thoughtful and constructive comments. We have followed the reviewer's suggestion to evaluate the prediction performance of AIDE-Q on the measurement data collected from noisy numerical simulations. The key finding is that **AiDE-Q demonstrates strong performance even in the presence of quantum noise, highlighting its robustness to low noise levels**. In the remainder of this reply, we separately introduce the construction of the training dataset for noisy quantum states, the hyperparameter setting, and the experimental results. All contents in this reply have been updated to the revised manuscript.
>
> $\bullet$ *Training dataset construction from noisy quantum states.* We first formulate the form of the training dataset when considering the quantum system noise. In particular, we employ the depolarization noise model $\mathcal{N}_q$ to characterize quantum system noise in the worst-case scenario. The prepared ground state under depolarization noise model $\mathcal{N}_q$ refers to
> $$\tilde{\rho}= \mathcal{N}_q[\rho] = (1-q)\rho + q \frac{\mathbb{I}}{2^N},$$
> where $0\le q\le 1$ refers to the noise rate, $\rho$ is the noiseless ground state corresponding to the case of $q=0$, and $\mathbb{I}/2^N$ represents the $N$-qubit maximally mixed state. In this regard, we denote the hybrid dataset for quantum property estimation (QPE) as
> $$\tilde{\mathcal{S}} _{\text{hyd}}=\tilde{\mathcal{S}} _{L}\cup \tilde{\mathcal{S}} _U, \text{with } \tilde{\mathcal{S}} _{L}=\\{ \tilde{\mathbf{x}}^{(i)}|\tilde{\mathbf{o}}^{(i)}\in\mathbb{R}^{Nm_l} \\} _{i=1}^{n_l}  \text{ and } \tilde{\mathcal{S}} _{U}=\\{\tilde{\mathbf{x}}^{(i)}|\tilde{\mathbf{o}}^{(i)}\in\mathbb{R}^{Nm_u}\\} _{i=n_l+1}^{n_l+n_u},$$
> where $\tilde{\mathbf{x}}^{(i)}=(\mathbf{p}^{(i)},\mathbf{M}^{(i)}, \tilde{\mathbf{o}}^{(i)})$ refers to the collected classical input about the noisy ground state $\tilde{\rho}(\mathbf{p}^{(i)})=\mathcal{N}_q[\rho(\mathbf{p}^{(i)})]$ with physical parameters $\mathbf{p}^{(i)}$, and $\tilde{\mathbf{o}}^{(i)}$ is the measurement output of the Pauli operator $\mathbf{M}^{(i)}$. Here, $m_l,m_u$ refer to the number of measurements with $m_u<m_l$. Let $\tilde{\mathbf{y}}^{(i)}$ be the noisy label of $\tilde{\mathbf{x}}^{(i)}$ with the noise coming from the statistical measurement error and the quantum system noise.
>
>
> $\bullet$ *Hyperparameter setting.* The experiment setup for evaluating the performance of AiDE-Q is as follows. The hybrid dataset $\tilde{S}_{\text{hyd}}$ is collected from noisy quantum states, focusing on the 10-qubit Heisenberg XXZ model. We set the noise rate of the depolarization noise model $\mathcal{N}_q$ as $q\in\\{0.1,0.3,0.5\\}$.
>
> The other hyperparameters are kept the same as those in our manuscript. In particular, the total number of training data is set as $n=720$, the ratio of the number of high-quality data $n_l$ to the total dataset size $n$ is defined as $r := n_l/n = 0.4$. The number of measurements for the initial high-quality dataset $\mathcal{S} _L$ is set to $m_l = 2^{10}$, while for the initial low-quality dataset $\mathcal{S} _U$, the number of measurements is $m_u \in \\{2^6, 2^7\\}$.
>
> $\bullet$ *Experimental results.* The table below presents the $\mathrm{R}^2$ values in predicting the entanglement entropy $S_A$ for the supervised learning (SL) models with and without incorporating AiDE-Q, corresponding to the labels 'SL' and 'SL w. DE'.
>
> | Method |      $m_u=2^6$ | | | | $m_u=2^7$ | | | |
> |:-------|:----------|:----------|:----------|:----------|:----------|:----------|:----------|:----------|
> | | $q=0$ | $q=0.1$ | $q=0.3$ | $q=0.5$ | $q=0$ | $q=0.1$ | $q=0.3$ | $q=0.5$ |
> | SL | 0.754 | 0.783 | 0.701 | 0.402 | 0.734 | 0.725 | 0.74 | 0.462 |
> | SL w. DE | 0.854 | 0.863 | 0.769 | 0.457 | 0.82 | 0.789 | 0.808 | 0.542 |
> | **Improvement** | **0.10** | *0.086* | 0.064 | 0.035 | **0.085** | 0.062 | 0.068 | *0.08* |
>
> **Note**: Best results are shown in **bold** and second-best results in *italics*.
>
> An immediate observation is that while the prediction performance of supervised learning models significantly decreases for a large noise rate $q=0.5$, **AiDE-Q could still effectively improve the prediction performance of learning models for various noise levels** $q\in \\{0.1,0.3,0.5\\}$. The improvement in $\mathrm{R}^2$ decreases from 0.1 for the noiseless case of $q=0$ to 0.035 for the noisy case with $q=0.5$, when the number of measurements is small for the low-quality data, namely $m_u=2^6$. On the other hand, for a larger number of measurements $m_u = 2^7$, the AiDE-Q could remain a relatively large improvement of $\mathrm{R}^2$, namely 0.08, even when the noise rate reaches $q=0.5$. This indicates that the improvement of prediction performance raised from AiDE-Q is **robust to the quantum system noise** for a large number of measurements $m_u$.
>
> **[W2]: I didn't really spot many weaknesses outside of a typo or two. ''Basic of quantum computing'' should be ''Basics of quantum computing''**
>
> **A2:** Thank you for this feedback. We will conduct a thorough proofreading of the entire manuscript and correct all typos in the revised version.

---

> > ### Comment · Reviewer_nKML · 2025-08-05
> >
> > Thank you for running additional demonstrations with noise. For what it is worth, depolarizing errors are likely not the worst-case scenario in a machine-learning set up. Typically ML algorithms instead struggle the most to handle quantum data that has been affected by coherent errors.
> >
> > Depolarizing errors act uniformly on states, which makes it easy for an ML algorithm to undo their effect. This result is generally true for Pauli stochastic errors as well. Coherent errors are harder because their affect on states is governed by unitary dynamics. The very poor analogy is that stochastic errors act classically, while coherent errors act quantumly.
> >
> > But the above has little impact on the validity of the demonstrations.

---

> > > ### Author Response · Authors · 2025-08-06
> > >
> > > Dear Reviewer nKML
> > >
> > > We thank the reviewer for their valuable feedback and insights, and appreciate their assessment of the overall validity of our results in the noisy scenario. We agree with the reviewer's point that coherent errors are more challenging for machine learning algorithms to address. We have included a discussion on the effects of coherent errors in the **Limitations** section of the revised manuscript and leave the exploration of their effects for future work.

---

### Decision · Program_Chairs · 2025-09-17

**Decision:**

Accept (poster)

**Comment:**

The manuscript addresses the problem of limited datasets when training deep learning models to estimate the ground state properties of quantum systems. The proposed approach is based on synthetic data with the main technical contribution being a method for consistency-check. The resulting algorithm has been evaluated in a series of experiments with up to 50 qubits. The main strengths are the high relevance, the thorough evaluation, and the good presentation, while a lack of theoretical analysis is the main weakness.
Only three reviews were provided, after rebuttal and discussion are all reviewers rating accept or BA. The reviewers basically agree on the strengths and weakness above, except for nKML who criticized a lack of noisy experiments instead of theoretical analysis..